# Nucleotide dependency analysis of genomic language models detects functional elements

Pedro Tomaz da Silva [1,2,8], Alexander Karollus[1,2,8], Johannes Hingerl [1,2], Gihanna Sta. Teresa Galindez[1,3], Nils Wagner [1], Xavier Hernandez-Alias [1,4], Danny Incarnato [5] & Julien Gagneur [1,6,7]✉

Deciphering how nucleotides in genomes encode regulatory instructions and molecular machines is a long-standing goal. Genomic language models (gLMs) implicitly capture functional elements and their organization from genomic sequences alone by modeling probabilities of each nucleotide given its sequence context. However, discovering functional genomic elements from gLMs has been challenging due to the lack of interpretable methods. Here we introduce nucleotide dependencies, which quantify how nucleotide substitutions at one genomic position affect the probabilities of nucleotides at other positions. We demonstrate that nucleotide dependencies are more effective at indicating the deleteriousness of genetic variants than alignment-based conservation and gLM reconstruction. Dependency analysis accurately detects regulatory motifs and highlights bases in contact within RNAs, including pseudoknots and tertiary structure contacts, revealing new, experimentally validated RNA structures. Finally, we leverage dependency maps to reveal critical limitations of several gLM architectures and training strategies. Altogether, nucleotide dependency analysis opens a new avenue for discovering and studying functional elements and their interactions in genomes.

The basic blueprint of every living organism is encoded in its genome. While high-throughput sequencing allows us to read this genetic information, interpreting its meaning remains a major challenge. A key interpretation method is sequence comparison[1], which identifies functional elements by leveraging nucleotide-level conservation as well as statistical dependencies between nucleotides. Covariation analysis, in particular, has been crucial in structural biology[2,3], for instance, in identifying conservation of Watson–Crick base pairing in RNA. However, these analyses traditionally relied on sequence alignments, limiting their use to highly conserved genomic regions.

Genomic language models (gLMs) have emerged as an alignment-free alternative[4,5]. Trained to predict nucleotides from their sequence context, these models learn evolutionary patterns directly from vast amounts of genomic data[4]. Studies have shown that gLMs capture biologically relevant information, distinguishing between functional and nonfunctional transcription factor (TF)

[1]School of Computation, Information and Technology, Technical University of Munich, Munich, Germany. [2]Munich Center for Machine Learning, Munich, Germany. [3]Munich Data Science Institute, Technical University of Munich, Munich, Germany. [4]Mechanisms of Protein Biogenesis, Max Planck Institute of Biochemistry, Martinsried, Germany. [5]Department of Molecular Genetics, Groningen Biomolecular Sciences and Biotechnology Institute (GBB), University of Groningen, Groningen, the Netherlands. [6]Institute of Human Genetics, School of Medicine and Health, Technical University of Munich, Munich, Germany. [7]Computational Health Center, Helmholtz Munich, Neuherberg, Germany. [8]These authors contributed equally: Pedro Tomaz da Silva, Alexander Karollus. ✉e-mail: gagneur@in.tum.de

**Fig. 1 | Probing nucleotide dependencies from gLMs. a**, gLMs are trained on genomes to predict nucleotides given their sequence context, assigning a probability to each one of A, C, G or T. **b**, We probe pairwise nucleotide dependencies from gLMs by quantifying how substituting a nucleotide at a query position affects predicted probabilities at a target position. **c**, Correlation between the absolute variant effect on gene expression, as measured using a saturation mutagenesis assay of 9 human promoters ($n = 8,635$ variants), and the variant influence score, the reconstruction score (log-likelihood ratio of substituting the reference to the alternative nucleotide according to the gLM), alignment-based conservation scores from PhyloP and PhastCons based on the 100-way, 447-way and 470-way alignment, the supervised model Borzoi, as well as a linear regression on the influence score and Borzoi's predicted absolute fold changes (the latter was fitted and evaluated using fivefold CV on this dataset). gLM log ratio quantifies the reconstruction at a specific position (as depicted in **a**), while the variant influence score quantifies how each variant affects the predicted probabilities across all target nucleotides in a sequence (this study, as depicted in **b**). The height of each bar corresponds to the average correlation across promoters between a score and the measured variant effect. Error bars

represent ±2 s.d., constructed using 100 bootstrap samples per promoter. **d**, Left: annotated nucleotide dependency map for the *S. cerevisiae* arginine transfer RNA, tR(ACG)O. The gray heatmap (top) shows log-odds ratios for all four nucleotides of a target (columns) when substituting the query nucleotide to each of the three alternatives (rows). These data are shown for the query being nucleotide 1 of the tRNA (T) and target being nucleotide 72 (A). The gLM log-odds ratios are consistent with the fact that these two bases encode a Watson–Crick contact in the RNA fold. The maximum absolute log-odds ratio, which defines the dependency score between those two positions, is realized when substituting an A on the query and having a T at the target. The dependency map (blue-to-red heatmap) shows dependency scores for all queries (rows) and targets (columns) in this locus. The colored rectangles in the dependency map highlight antiparallel dependencies belonging to each of the tRNA arms, while the red square delineates a dependency between two bases in different loops of the tRNA contributing to its tertiary structure (red bases in the tertiary structure). The track above the dependency map displays the nucleotide reconstruction predicted by the gLM. Right: annotated tertiary structure of tR(ACG)O[48]. CV, cross-validation.

binding motifs and identifying genetic variants with phenotypic effects[4–6]. They have also found use as so-called foundation models for predicting molecular phenotypes, sometimes outperforming other methods[4,7–16]. These analyses indicate that gLMs intrinsically represent genomic functional elements. However, the foundation model paradigm uses gLMs as intermediate black boxes and does not reveal these elements.

In this work, we leverage gLMs to provide a measure of dependencies between nucleotide pairs. We systematically study the resulting nucleotide dependency maps to determine which genomic elements they encode and thus exploit them to characterize functional elements and their interactions. This approach also allows us to compare different gLMs and identify their limitations.

## Results

### Nucleotide dependency maps

Genomic language models are trained to reconstruct nucleotides, thereby providing nucleotide probabilities given their surrounding sequence context (Fig. 1a). In principle, success at reconstructing nucleotides requires detecting characteristic genomic features that are more likely to be found in the sequence context. For example, the probability of a particular nucleotide in the human genome being a guanine strongly depends on whether it is intronic (~22% (ref. 17)) or located at the third base of a start codon (~100%). To study the relationship between nucleotides and their context using gLMs, we use a technique analogous to in silico mutagenesis (explained in ref. 18). Specifically, we mutate a nucleotide in the sequence context (query nucleotide) into all three possible alternatives and record the change in predicted probabilities at a target nucleotide in terms of odds ratios (Fig. 1b and Methods). This procedure, which can be repeated for all possible query-target combinations, quantifies the extent to which the language model prediction of the target nucleotide depends on the query nucleotide, all else equal.

We applied this general procedure to 14 gLMs (Extended Data Table 1 and Methods). Unless stated otherwise, we present results from our SpeciesLM gLMs that were trained on regions 5′ of start codons in fungi and metazoa (SpeciesLM fungi and SpeciesLM metazoa; Methods). On selected biological applications, we turn to other gLMs. To assess the biological relevance of these dependencies, we sought to verify that single-nucleotide variants (SNVs) of known functional importance have a greater impact on gLM predictions. Given an SNV at a query position, we computed for any target position the maximum absolute log-odds ratio over all possible four target nucleotide values. Next, we averaged these values across all targets to obtain an aggregate score of query variant impact (Methods). We named this metric the variant influence score. In the ClinVar database[19], the influence score was significantly higher for noncoding pathogenic variants than for benign variants (Extended Data Fig. 1a,b). This is despite using a gLM trained only on 2-kb regions 5′ of start codons, which only overlap a small fraction of all transcribed bases. Prior studies leveraged gLM reconstruction probabilities to prioritize functional variants,

positing that lower probability indicates greater deleteriousness[5,6] (Methods). Remarkably, this reconstruction-based metric showed substantially lower performance than the influence score. However, the influence score did not outperform alignment-based scores, perhaps because the criteria used by ClinVar to categorize variants as pathogenic include bioinformatics predictions that often integrate alignment-based conservation.

For a less biased comparison, we focused on a dataset from a saturation mutagenesis experiment on nine selected human promoters[20] (Fig. 1c). Here the variant influence score correlated with variant effects on absolute gene expression fold change, outperforming reconstruction, as well as alignment-based conservation scores[21–24]. Remarkably, the purely unsupervised influence score was on par with the state-of-the-art supervised expression predictor Borzoi[25]. These two approaches appeared to capture complementary predictive signals, because a simple integrative model further improved performance (Fig. 1c; similar observations on noncoding ClinVar variants are shown in Extended Data Fig. 1b). The variant influence score also outperformed reconstruction and alignment-based conservation at distinguishing fine-mapped promoter expression quantitative trait loci (eQTLs) single-nucleotide polymorphisms from matched controls, in human, where it did not outperform Borzoi, and in yeast[26–29] (Extended Data Fig. 1c–f).

Having shown that aggregate dependency strengths reflect functional importance, we then studied individual query-target pairs. For every query-target pair, we considered the maximum effect a query nucleotide change has on the predicted odds of a target, yielding two-dimensional (2D) nucleotide dependency maps (Methods). An example map is shown for the yeast arginine tRNA (Fig. 1d). The entire secondary structure of the tRNA, defined by base pairing within the four arms, clearly stands out with high dependencies. The dependency map also highlighted a tertiary structure contact. Upon introducing single-nucleotide substitutions in these pairs, the gLM adapted its predictions according to the Watson–Crick base pairing and, with a lesser preference, to wobble base pairing (see Fig. 1d for an example). Remarkably, the model recapped structural RNA rules from its reconstruction objective alone, in an alignment-free manner and without focused training on tRNAs.

Nucleotide pairs have two dependencies, depending on which nucleotide is the query. Scoring nucleotide pairs by the maximum of those two values yielded near-perfect secondary structure contact predictions across 172 tRNAs of *Saccharomyces cerevisiae*. Alternative metrics, including gradient-based dependencies and using masking instead of nucleotide substitution on query, showed lower predictive signal. This trend was confirmed when further assessing the dependencies on cognate donor and acceptor splice sites (Extended Data Fig. 1g,h).

In the following sections, we explore and categorize patterns found in nucleotide dependency maps, associate them to biological mechanisms and exploit them to detect and characterize functional elements in the genome.

**Fig. 2 | Blocks along the diagonal of dependency maps highlight instances of regulatory sequence motifs. a**, SpeciesLM fungi nucleotide reconstructions (scaled by information content) and nucleotide dependency map for the *SMT3* promoter (yeast). TF motifs and poly(dA:dT) are reconstructed with similar confidence, whereas blocks appear only for TF motifs in the dependency map. Ground-truth motifs from YeTFaSCo. **b**, Examples of dependency blocks from human promoters. From top to bottom: Znf652 motif in the *LDLR* promoter, Nfy motif in the promoter of *MTO1* and a Spdef motif in the *OGA* promoter. Ground-truth motifs from Hocomoco v12 (ref. 49). **c**, Top: per nucleotide block scores for nucleotides in repeats, as marked by RepeatMasker[50] and those reported to be in a bound TF motif. The block score is computed as the first quartile of dependencies among consecutive spans of six nucleotides. Bottom: per nucleotide information content of the gLM reconstruction in

repeats and reported to be in a bound TF motif. For each boxplot: centerline, median; box limits, first and third quartiles; whiskers span all data within 1.5× interquartile ranges of the lower and upper quartiles. ***$P < 0.0001$, two-sided Wilcoxon rank-sum test. **d**, ROC curve comparing the ability of different metrics to classify whether a nucleotide is part of a bound TF motif or not (92,117 binding nucleotides of 6,538,427 overall). The dependency block score performs substantially better than using the gLM nucleotide predictions and is comparable to yeast expert PWM scanning. This is despite the fact that PWMs were derived from in vitro and in vivo binding assays and were used to define the positive class, whereas the language model has never been exposed to binding data during training. **e**, Dependency map for an instance of the yeast Abf1 spaced motif, compared to the ground-truth binding preference from YeTFaSCo[30]. TPR, true positive rate; FPR, false positive rate.

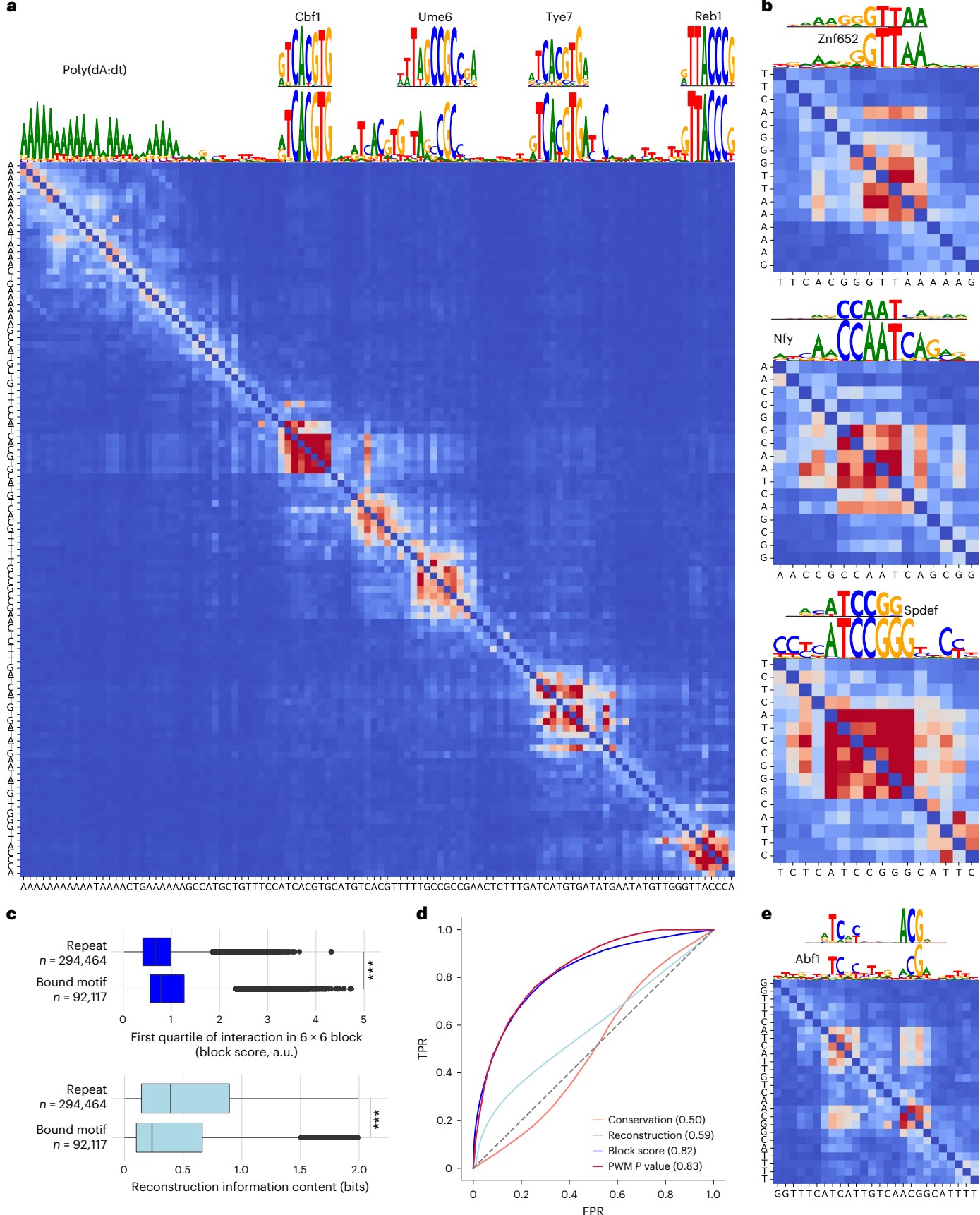

## Blocks along the diagonal highlight regulatory sequence motif instances

We observed that short sets of contiguous nucleotides frequently exhibited strong reciprocal dependencies, manifesting as dense blocks along the diagonal. Many dense blocks were observed at TF motif instances in promoters (Fig. 2a,b). This was in striking contrast to other well-reconstructed locations, including simple repeats such as poly(dA:dT) stretches. Intuitively, an individual mutation in the poly(dA:dT) stretch will have a mild impact on predicting any other element of the stretch and thus dependencies in the repeat tend to be less dense. In contrast, all bases in a TF motif are strongly interdependent, as a mutation at any position can disrupt the entire site's function by reducing binding. Therefore, we reasoned that TF motifs could be detected using gLMs by searching for dependency blocks.

To find dependency blocks, we computed the first quartile of query-target dependencies among consecutive six nucleotides (Methods). This quantile-based block score is more robust than the average in isolating strong interactions, while privileging dense blocks. To assess how block scores facilitate the detection of TF binding sites, we leveraged the near-complete TF binding data in *S. cerevisiae* with nucleotide-level preferences (position weight matrices (PWMs)[30]). We considered the 1-kb regions 5′ of start codons and defined PWM matches within 10 bp of an experimental binding peak as binding sites for 68 TFs[31].

While reconstruction varied widely for binding site nucleotides and repeat elements, the block score of binding site nucleotides was generally higher than for nucleotides in repeats identified by Repeat-Masker (LTR retrotransposons, telomeric/centromeric repeats, rDNA regions, low-complexity DNA, simple repeats; Fig. 2c). Consistent with this, the block score discriminated binding site nucleotides substantially better than reconstruction (Fig. 2d). By comparison, the PhastCons conservation inferred from the alignment of seven yeast *Saccharomyces* species[21] had no discriminative power at this task, and modest discriminative power if overlapping coding sequences were removed (Fig. 2d and Extended Data Fig. 2). Extending the alignment to 69 species of the Saccharomycetales order using default parameters did not improve results (Extended Data Fig. 2 and Methods).

Moreover, the block score discriminated binding site nucleotides as effectively as PWM scanning. This result is remarkable because the block score was obtained in a completely unsupervised fashion, whereas the PWMs were not only derived from experimental data but also used to define the positive class. Additionally, if we only benchmark on nucleotides forming part of a PWM match, the block score will demonstrate an ability to discriminate binding from nonbinding PWM matches, thus showing that the gLM considers the context of the motif (Extended Data Table 2). In sum, this analysis demonstrates the ability of gLMs to detect regulatory elements and the utility of dependency maps.

We note that not all motifs appear as complete blocks. *S. cerevisiae* Abf1 motif, for example, is represented as two spaced and interacting blocks, reflecting the dimeric binding preferences of this factor (Fig. 2e). Thus, even within motifs, the dependency maps can serve to visualize underlying functional relationships.

## Off-diagonal blocks indicate sequence element interactions

Blocks in the dependency maps also occurred away from the diagonal, revealing distal interactions, such as between key transcription initiation elements (TATA box and INR) in *Drosophila melanogaster* (Fig. 3a) and the primary splicing determinants (donor, branch and acceptor sites) in *S. cerevisiae* (Fig. 3b and Extended Data Fig. 3). The short length of yeast introns allowed a genome-wide assessment that showed that dependencies between donor and acceptor splice sites were higher than dependencies between donor and decoy acceptor-like sequences within the intron or background dependencies at matched distances (Fig. 3c). These results indicate that distal dependencies capture a range of functional relationships among sequence elements, including promoter and transcript architecture.

Going a step further, we asked whether the maps could also reflect changes in transcript structure due to interindividual variation. To this end, we leveraged aberrant splicing events associated with rare variants from 946 human individuals (GTEx[32]) and SpliceBERT, a language model trained on vertebrate RNA sequences[14]. As an example, a rare variant in the *TRPC6* gene disrupts a canonical donor splice site, leading to the use of a cryptic site and the creation of an aberrant, shorter intron. The dependency map reflects this by showing a strong interaction between the canonical donor and the boundaries of this new intron (Fig. 3d). Across 1,811 rare-variant-associated aberrant splicing events, dependencies between the variant position and the ends of the corresponding outlier intron exceeded those between nucleotides at matched distances (Fig. 3e). These results held for both outlier intron ends and all variant location categories (Fig. 3e). We conclude that dependency maps capture splicing rules and can reflect variant-induced transcript structure alterations.

## Nucleotide dependencies reveal RNA secondary and tertiary structure contacts

Besides blocks, we observed antiparallel diagonals, that is, distal stretches of consecutive nucleotides that depend on each other one-to-one in reverse order as in the case of the four arms of the yeast arginine tRNA described above (Fig. 1d). Using a convolutional filter, we systematically called regions with antiparallel elements across diverse fungal genomes (Methods and Extended Data Fig. 4a). Dependencies in antiparallel diagonals were typically consistent with Watson–Crick or wobble base pairing (Extended Data Fig. 4b), indicating that they captured helical stems, critical to RNA folding. Moreover, antiparallel diagonals with the strongest dependencies were found

**Fig. 3 | Off-diagonal blocks highlight sequence element interactions.**
**a**, Dependency map extracted from the SpeciesLM metazoa in the promoter of the *D. melanogaster* gene *GstO2*. On top is the reconstruction (scaled by the information content) from the gLM, highlighting the TATA box and initiator (INR) element motifs. High dependencies can be spotted at their intersection in the dependency map, reflecting their functional interaction. **b**, Dependency map for the intron of the yeast gene *ATG44* together with exonic flanking regions of eight nucleotides. The top track corresponds to the nucleotide reconstruction (scaled by the information content) that highlights the donor and branch point motifs. Off-diagonal dependencies can be spotted in the intersection between these motifs and the acceptor, indicating their interdependence. **c**, Average dependency between donor and acceptor nucleotides; donor and acceptor-like decoy nucleotides (AG dinucleotides within the intron not part of an annotated 3′ intron end); donor and random nucleotide pairs matching donor–acceptor distances. \*\*\*P < 0.0001, two-sided Wilcoxon rank-sum test. **d**, Exon elongation variant in a human individual on an intron of the gene *TRPC6*. Top: sashimi plots for an individual without the variant (top track) and an individual with the variant (bottom track), indicating differential splicing (number of RNA-seq reads supporting each splice junction) resulting from a variant-inducing exon elongation. Bottom: dependency map obtained from SpliceBERT showing a dependency between the canonical and alternative (alt.) donor, indicating that a substitution in the canonical donor site induces a change in predicted probability for the alt. donor position shown in the sashimi plots. **e**, For each variant location with respect to the splice site, each boxplot shows the average dependency between a variant position and its reported outlier junction donor or acceptor, and average dependencies for nucleotides at distances matching the spacing between the variant and the outlier donor or acceptor. Splice acceptor, n = 45; splice donor, n = 69; splice region, n = 147; exon, n = 736; intron, n = 814. All comparisons between dependencies for donor and acceptor and dependencies at matched distances were significant (one-sided Wilcoxon rank-sum test, all P < 10^-12). For each boxplot: centerline, median; box limits, first and third quartiles; whiskers span all data within 1.5× interquartile ranges of the lower and upper quartiles.

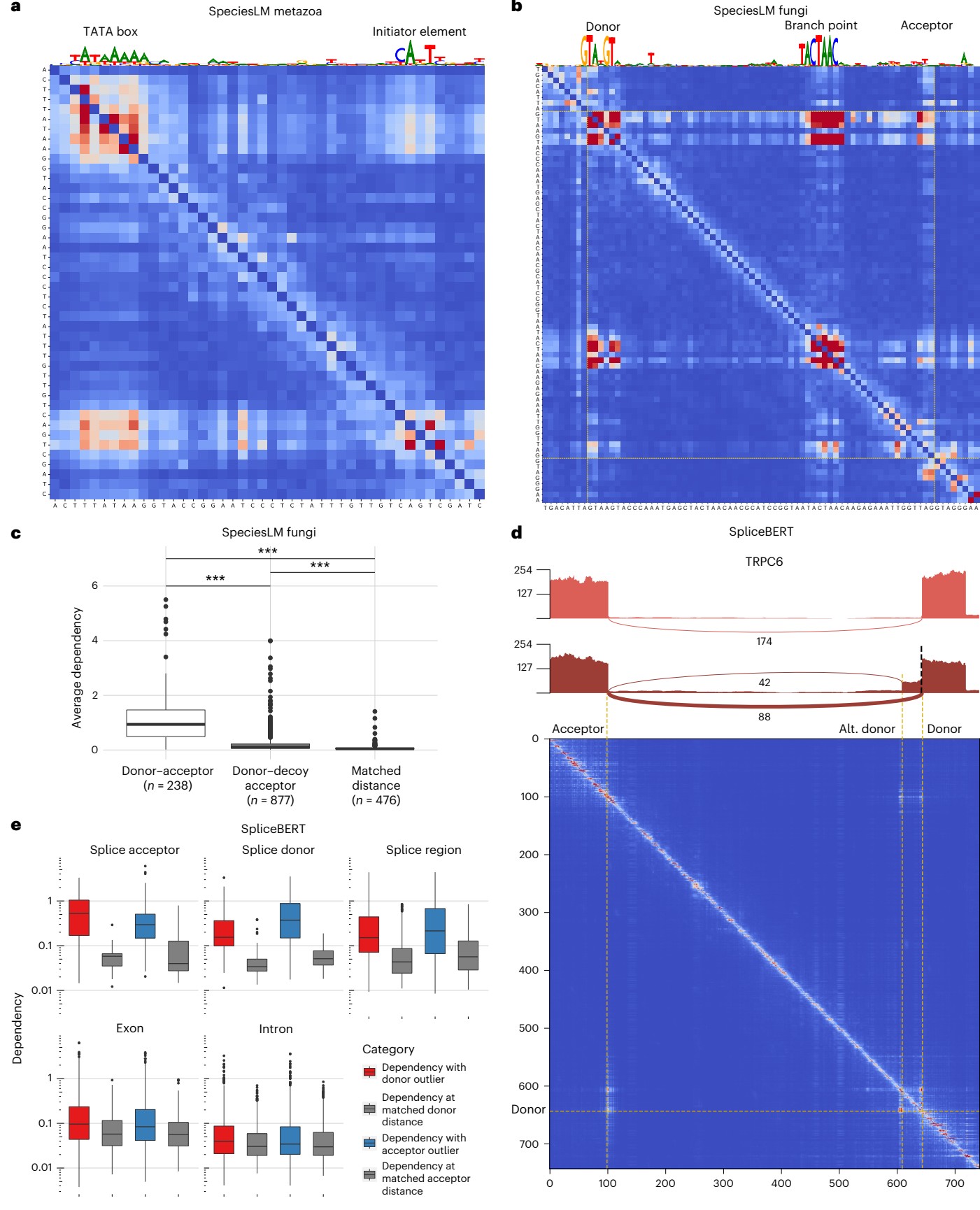

among highly structured RNAs such as tRNAs and ribosomal RNAs (Extended Data Fig. 4c and Methods). Hence, these findings suggest that detecting antiparallel diagonals in nucleotide dependency maps could be instrumental in inferring RNA structures.

To evaluate the potential of dependency maps for capturing RNA structures, we used RiNALMo, a language model trained on 36 million noncoding of both annotated and unannotated RNA sequences from RNAcentral[33], nt[34], Rfam[35] and Ensembl[36], spanning a wide variety of species[7]. Originally, the authors of RiNALMo trained this LM as foundation for a supervised predictor of RNA secondary structures. We, instead, scored contacts as the largest of the two dependency map entries for each pair of nucleotides, computed using RiNALMo's underlying LM. Remarkably, these scores were strongly predictive of secondary structure contacts, with areas under the receiver operating characteristic (ROC) curve typically exceeding 0.9 for most RNA families (Archive II database[37]; Fig. 4a), although we performed no fine-tuning on secondary structures. Nonetheless, tools optimized for RNA secondary structure analysis, such as RNAalifold and the fine-tuned RiNALMo, outperformed dependency scores at predicting secondary structure contacts of experimentally validated structures (Extended Data Fig. 4d).

However, secondary structures are simplified planar representations of the topology of a single possible 3D folding of an RNA sequence, missing important contacts occurring in the 3D fold. We observed that some apparent false positive predictions from our dependency-map approach corresponded to tertiary structure contacts that were absent from predictions by the supervised RiNALMo. For instance, in *Archaeoglobus fulgidus* isoleucine tRNA, the dependency maps showed 6 bp with dependencies as strong as secondary-structure contact dependencies (dependency > 6). These constituted six of the eight known contacts found only in the tertiary structure. RiNALMo's supervised secondary structure predictor unsurprisingly missed these tertiary structure contacts (Fig. 4b). This ability to detect tertiary interactions is important, as they provide useful spatial constraints to help determine an RNA's 3D structure.

To systematically evaluate the added value of dependency maps in capturing tertiary structure contacts, we analyzed noncanonical (that is, non-Watson–Crick/wobble) contacts in Protein Data Bank (PDB) RNA structures in the CompaRNA database[38]. We found that 50% of the pairs with a dependency score larger than 13.5 and not predicted to be in secondary structure by RiNALMo's supervised model were annotated as contacts (Fig. 4c). Across the entire database, noncanonical base pairs were well-captured by the dependency maps (Fig. 4d; area under the curve (AUC) = 0.8). In contrast, this information was largely lost by the supervised RINALMo and by RNAalifold (Fig. 4d and Extended Data Fig. 4d). These results indicate the utility of dependency maps for RNA structure inference by providing candidate contacts not captured by secondary structure contact predictors.

These findings prompted us to investigate further the potential of dependency maps in addressing major challenges of secondary structure prediction, such as pseudoknot detection. Pseudoknots are important nonsecondary structure elements that form when base

pairs are not nested, for example, bases in a loop pairing with another single-stranded region. We observed high dependencies between bases of documented contacts implied by pseudoknots, such as in the 396 nt-long RNase P RNA (see Fig. 4e and Extended Data Fig. 4e for another example in a riboswitch), in which not only the stems but also the pseudoknot are reflected with strong antiparallel diagonals. Analyzing systematically 2,530 pseudoknot-containing RNA structures with less than 90% sequence similarity from the bpRNA-1m(90) dataset[39], we found that pairs of nucleotides in pseudoknots showed substantially higher dependencies than pairs not belonging to structural contacts (AUC = 0.92; Extended Data Fig. 4f).

An RNA's secondary structure represents the topology of a single conformation. However, an RNA sequence can adopt alternative functional RNA folds. We found that dependency maps can capture alternative structures. For instance, the dependency maps of the tryptophan leader sequence in the bacterium *Escherichia coli* (Fig. 4f), a structured region for which tryptophan abundance regulates the switch between terminator and antiterminator conformations[40], captures the two alternative folds, with domain 3 being involved in antiparallel diagonals with both domain 2 and domain 4 (ref. 40).

To assess the capacity of dependency maps to derive new structural predictions, we performed in-cell chemical probing of *E. coli* with DMS, followed by high-throughput mutational profiling analysis (DMS-MaPseq), a transcriptome-wide assay probing adenines and cytosines not engaged in Watson–Crick base pairing[41]. Transcriptome-wide, the structural contacts predicted by the antiparallel patterns in dependency maps can efficiently capture experimentally derived RNA base-pair contacts (Extended Data Fig. 4g,h).

We then focused on all noncoding regions upstream of the start codon spanning 500 nucleotides, as they harbor different transcribed regions, including structures with roles in translation and transcription regulation[42]. We selected dependency maps indicating the presence of at least two stem loops and not belonging to an annotated structure, revealing four previously unreported secondary structures corroborated by experimental data from DMS-MaPseq and validated by covariation analysis (Fig. 4g and Extended Data Fig. 4i). Notably, as covariation analysis typically requires a high-quality sequence alignment and, optionally, a predicted RNA structure, the ability of nucleotide dependencies to capture—in an alignment-free and unsupervised fashion—functionally relevant RNA structural contacts, underscores their predictive power.

Collectively, these results show that dependency-map analysis can overcome the typical challenges associated with RNA structure prediction, capturing both secondary and tertiary structure contacts, pseudoknots and alternative structures of functionally relevant RNAs.

## gLMs capture forward and inverted duplications without memorization

We observed parallel (Fig. 5a and Extended Data Fig. 5a) and antiparallel diagonals reflecting duplicated sequences in the forward and reverse complement orientations, respectively. Further in silico experiments

**Fig. 4 | Dependency maps reveal known and new RNA structures, and highlight tertiary contacts. a**, AUROC curve for the classification of RNA structure contact pairs from the Archive II dataset spanning nine different RNA families. For each boxplot: centerline, median; box limits, first and third quartiles; whiskers span all data within 1.5× interquartile ranges of the lower and upper quartiles. **b**, *A. fulgidus* tRNA(Ile2) (PDB ID: 3AMU) dependency map (left), ground-truth contacts (middle) and contacts predicted by the fine-tuned RiNALMo (right). **c**, Ratio of correctly retrieved contacts (precision) not predicted by the supervised RiNALMo (predicted probability < 0.5) for each dependency value threshold. **d**, ROC curve for the classification of noncanonical structure contacts across the CompaRNA dataset showing that dependency maps capture non-Watson–Crick and tertiary structure contacts that are lost on the supervised RiNALMo (AUC = 0.64, $P < 10^{-4}$, permutation test). Canonical contacts, $n = 3873$;

noncanonical contacts, $n = 1762$. **e**, Left: *Bacillus subtilis* RNAse P secondary structure highlighting pseudoknot contacts. Right: corresponding dependency map showing antiparallel dependencies belonging to RNA structure stems and the annotated pseudoknot contacts. The structure was taken from the RFAM database[35] (ID: RF00011). **f**, Tryptophan operon leader dependency map together with annotation and representation of the secondary structure stems belonging to sequence domains 2, 3 and 4. **g**, Left: dependency map computed with RiNALMo of a region including 200-bp upstream of the gene *FkpB*. Right: DMS-MaPseq-derived secondary structure together with reactivities per nucleotide. The DMS-MaPseq data are consistent with the dependency map. The main structural features are highlighted by boxes. Each stem-loop is identified starting with 'H', and the pseudoknot with 'PK'. This structure was undescribed so far. PK, putative pseudoknot.

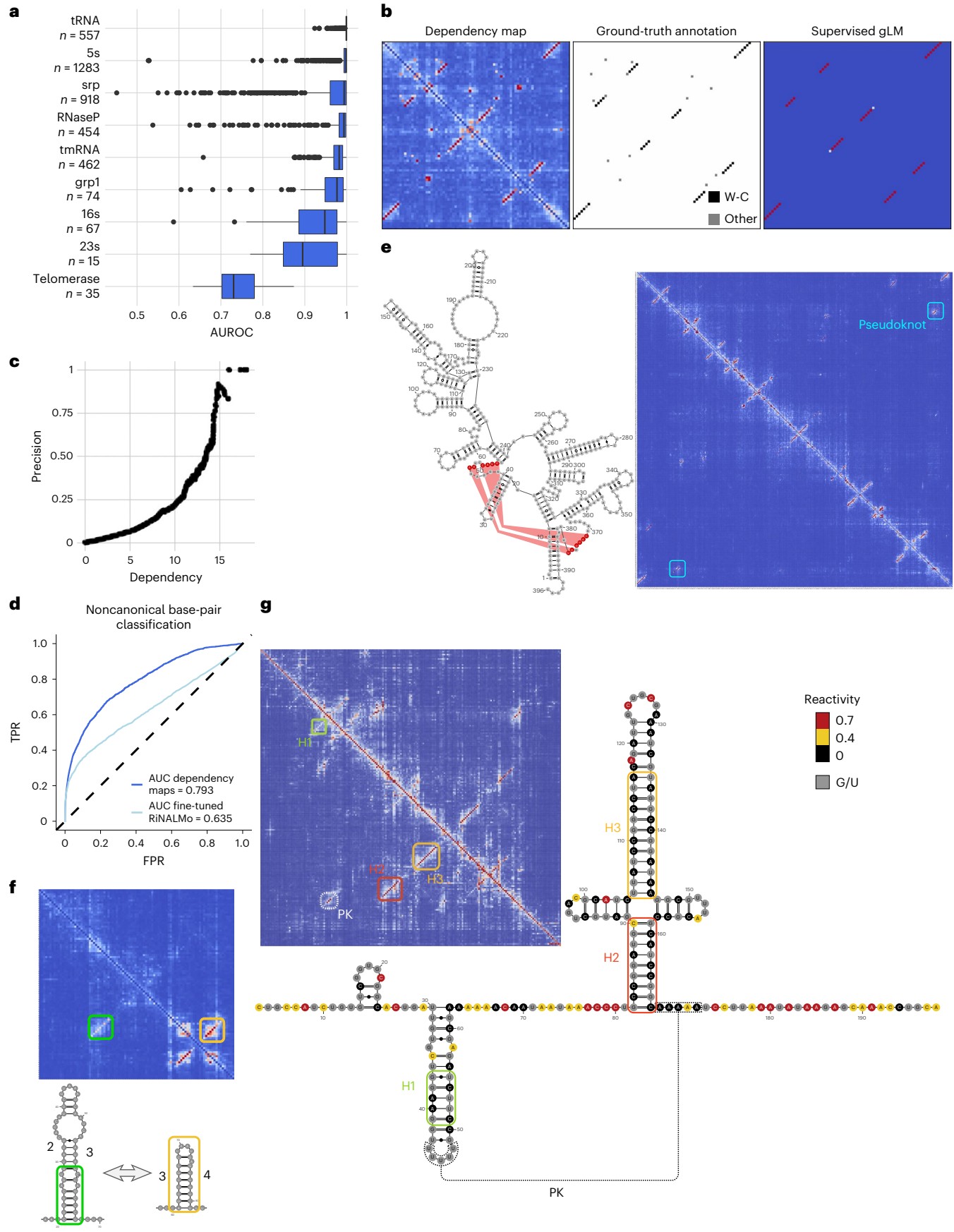

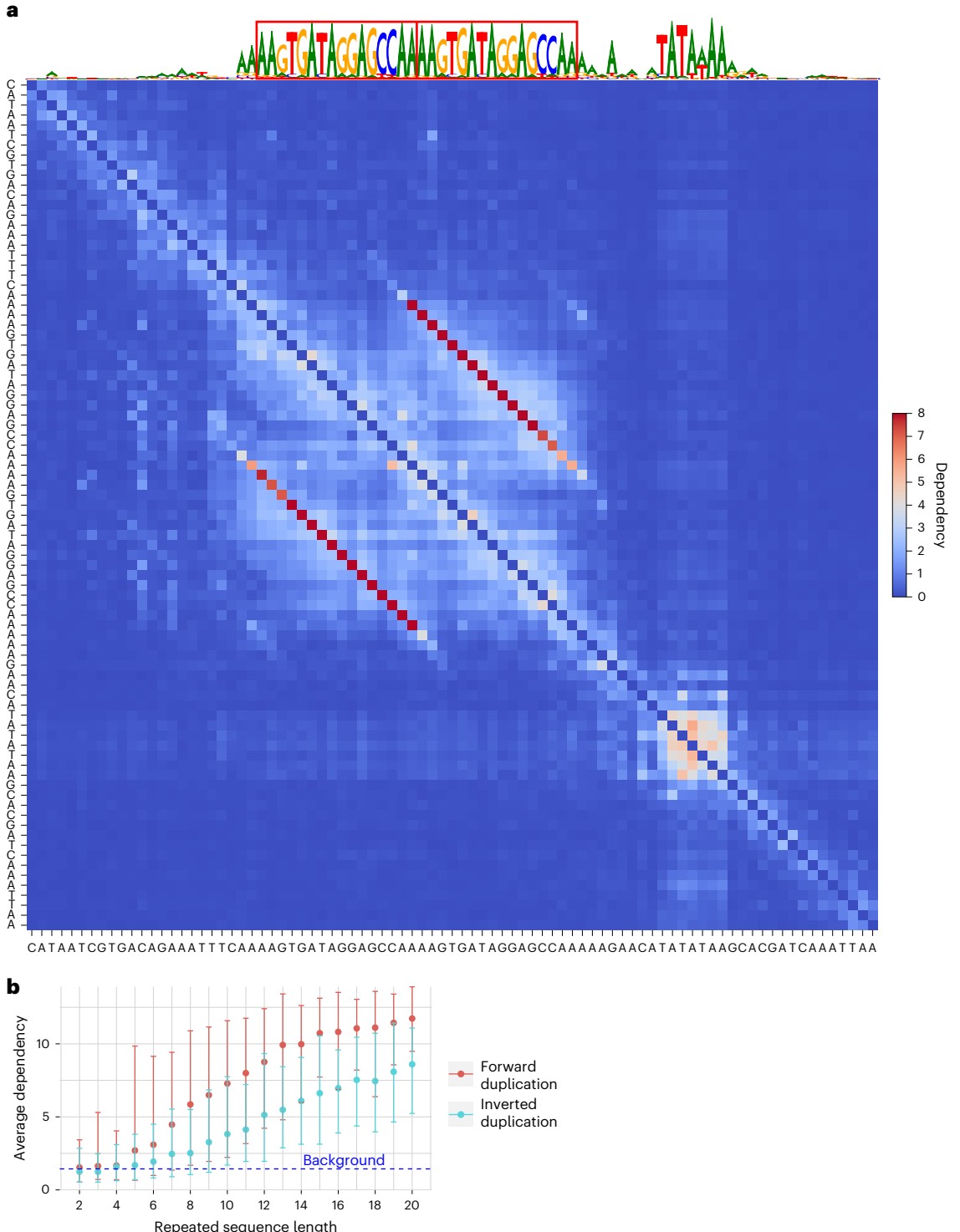

**Fig. 5 | gLMs capture sequence duplications without memorization.**
**a**, Dependency map from the SpeciesLM fungi in the promoter of *YNR064C* that contains a TATA box motif and duplicated sequences highlighted as red boxes on top of the reconstruction. While the TATA box appears as a block-like dependency pattern, the repeat shows a parallel dependency linking each duplicated nucleotide. **b**, Average dependency between artificially inserted repeat elements against their length for forward and inverted duplicates together (error bars represent 95% confidence interval computed across 100 samples).

demonstrated that gLMs have modeled the duplication operation itself, rather than relying on memorizing these sequences (Fig. 5b, Extended Data Fig. 5b and Supplementary Note).

Additionally, gLMs will only introduce short stretches of antiparallel dependencies in specific contexts, rather than associating any pair that could theoretically engage in Watson–Crick base pairing, demonstrating that the models have learned determinants of RNA

structure beyond reverse complementarity (Extended Data Fig. 5c and Supplementary Note).

**Dependency strength depends on genomic distance**
We then investigated pattern-independent, global properties of the distribution of dependencies. To this end, we focused on *S. cerevisiae* as a model system. Nucleotide dependencies followed a power–law

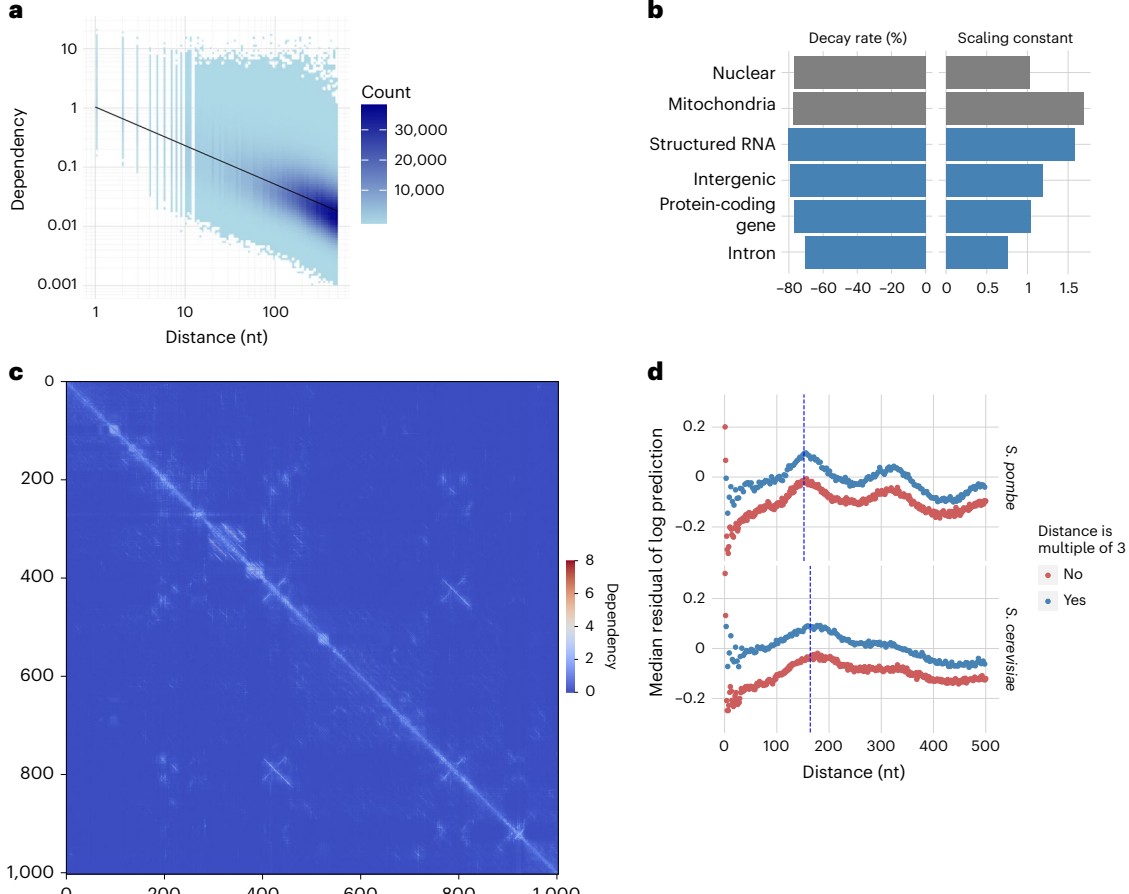

**Fig. 6 | Dependencies relate to distance in a species and region-specific way and reveal periodicities intrinsic to the genome. a**, Nucleotide dependencies computed from the SpeciesLM fungi against distance to the query nucleotide taken from dependency maps across the yeast genome. Linear regression fit (black line) marks the power–law relationship between dependency and distance. **b**, Power–law decay rate in percentage decrease in dependency per tenfold distance increase (left) and scaling constant (right) for different genomic regions (see Methods for exact category definitions). **c**, Example of a dependency map for the mitochondrial region 1 kb 5′ of gene *Q0017*. **d**, Median of the residuals of the fitted power law against distance to the query nucleotide for yeasts *S. pombe* and *S. cerevisiae*. The colors highlight the difference in dependency between targets distanced from the query by multiples of three nucleotides. The dashed blue lines show the nucleosome periodicity values reported for *S. cerevisiae* (164 bp) and *S. pombe* (152 bp)[43] and coincide with the highest deviations from the fitted power law, indicating that dependencies reveal and are constrained by nucleosome periodicity.

relationship with respect to distance to the query nucleotide, decaying by about 78% per tenfold distance increase (Fig. 6a). We did not find substantial variations in the decay rate across various types of genomic regions (Fig. 6b). However, dependencies were generally 1.64× stronger in the mitochondrial than in the nuclear genome (Fig. 6b). Browsing dependency maps of mitochondria revealed dependency-rich regions whose biological interpretation needs further investigations (Fig. 6c). Investigating deviations to the general power–law trend revealed higher dependencies at 3-nucleotide spacing, perhaps as a consequence of the high content of coding sequences in yeast. Nucleosome positioning also appeared to influence dependency distributions, with stronger dependencies than expected by the power law at distances corresponding to nucleosome position periodicity on both *S. cerevisiae* (164 bp) and *Schizosaccharomyces pombe* (152 bp)[43] (Fig. 6d). We conclude that nucleotide dependency maps offer new avenues to study general constraints on genomic sequences.

## Dependency maps uncover shortcomings in gLM model designs and training data selection

Current gLMs differ in both model architecture and the sequence data on which they were trained. As of the time of writing, there is no consensus on the advantages and disadvantages of these different approaches, and comparisons are challenging due to the complexity of gLMs. We set out to use nucleotide dependencies, which can be computed for any gLM, as a general tool for visualizing and getting insights into existing gLMs.

Human tRNAs are suitable loci for comparative analysis both because several models have been trained on human genomes only and because tRNAs entail well-established and highly conserved distal functional dependencies. We observed that some modeling choices introduce artifacts in the dependency maps. For example, models belonging to the Nucleotide Transformer family[9] do not reconstruct at the single base level but instead predict nonoverlapping spans of six nucleotides. This produces artificial dependency blocks along the diagonal, which do not represent motif instances but arise because nucleotides of the same span are generally more dependent (Fig. 7a). Nevertheless, these models are capable of learning dependencies at the single base level, for example, some tRNA stem contacts in the human *tRNA-Arg-TCT-4-1* (Fig. 7a).

Equally, autoregressive models, for example, Evo[8], do not consider bidirectional context when making predictions; instead, they are designed to predict the next nucleotide given its 5′ context. This creates an artifact at the beginning of genomic elements such as the tRNA, which likely arises because the model cannot deduce the element until it has seen sufficiently many tokens inside of it (Fig. 7a). This problem can be mitigated by running the model both on the forward

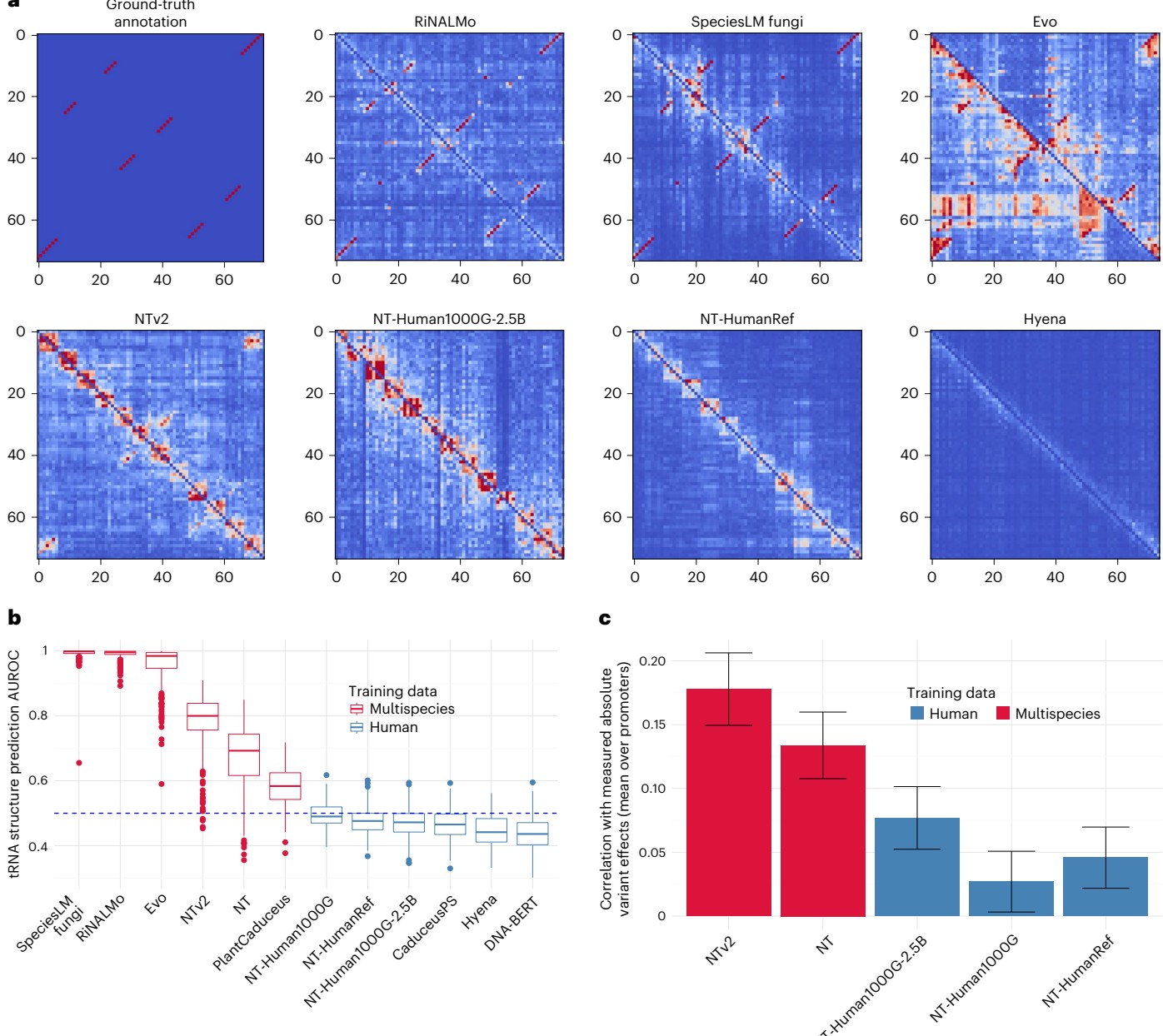

**Fig. 7 | Dependency maps to compare gLMs and diagnose their shortcomings.**
**a**, The top-left map shows the ground-truth tRNA secondary structure contacts taken from GtRNAdb (*tRNA-Arg-TCT-4-1*). Red indicates contact, and blue indicates no contact. The remaining maps show the dependency maps for different gLMs, revealing modeling artifacts and performance differences. For instance, Nucleotide Transformer version 2 (ref. 9) captures only a few of the structural interactions. The 6 × 6 blocks around the diagonal reveal an artifact of NTv2 nonoverlapping 6-mer tokenization. Evo[8] shows artifacts when encountering the start of genomic elements (see the top-left and bottom-right corners). Computing the maximum across the top and bottom diagonal can mitigate the artifact. **b**, Comparing the AUROC curve achieved when dependency maps, as computed using different models, are used to predict secondary structure contacts of 266 human tRNAs without fine-tuning. Models

differ in terms of architecture and training data. Multispecies models strongly outperform those trained only on the human genome, even if the multispecies models have never seen human (or even metazoan) DNA. For each boxplot: centerline, median; box limits, first and third quartiles; whiskers span all data within 1.5× interquartile ranges of the lower and upper quartiles. **c**, Comparing the correlation of the variant influence score, calculated using different Nucleotide Transformer models, with the measured absolute log fold change variant effect (Fig. 1c). Multispecies models perform substantially better than those trained only on human sequences, even for models of the 1000G type that were exclusively trained across 3,202 diverse human genomes. The height of each bar corresponds to the average correlation across promoters between a score and the measured variant effect. Error bars represent ±2 s.d., constructed using 100 bootstrap samples per promoter.

and reverse strand and taking the maximum dependency within a pair of nucleotides. Nevertheless, more appropriate measures of nucleotide dependencies for autoregressive models may need to be developed.

Comparing models trained on different types of sequences revealed starker differences. Specifically, models trained only on

the human genome, regardless of architecture, parameter count or whether within-species variation was included, did not learn the human tRNA structure to any meaningful degree (Fig. 7b). By contrast, models trained on multiple species succeeded in at least learning aspects of human tRNA structure, regardless of architecture and whether

the training data included any human genomes. Similar results were observed when evaluating the performance of gLMs from the Nucleotide Transformer family, which all show very similar architectures, on the human promoter saturation mutagenesis assay[20] (Fig. 7c) and ClinVar[19] (Extended Data Fig. 6). We conclude that infrequent genomic elements, even if they are highly conserved, generally require a multispecies approach to be learned.

## Discussion

In conclusion, we introduced nucleotide dependencies that quantify how nucleotide substitutions at one genomic position affect the likelihood of nucleotides at another position. This new metric appears as a general and effective approach to identifying functionally related nucleotides using gLMs. Nucleotide dependency maps reveal functional elements across various biological processes, including transcriptional, post-transcriptional regulatory elements, their interactions and RNA folding. Therefore, this new metric has implications across multiple areas of computational and genome biology.

Traditionally, comparative genomics has helped identify functional sequences by leveraging the concept of sequence conservation, a major indicator of functional importance based on purifying selection among homologous sequences, that is, sequences descended from a common ancestor. Algorithmically, sequence alignment is first used to identify homologous sequences; conservation is then estimated from the aligned nucleotide frequencies adjusted for phylogenetic drift and mutational biases. This approach limits the scope to alignable homologous sequences. In contrast, gLMs can more flexibly borrow information across sequences with similar contexts, allowing them to capture recurrent patterns such as TF binding site motifs and their functional arrangements that can have arisen independently on nonhomologous sequences. In principle, this also allows gLMs to capture instances of positive selection, for example, where a sequence element has been acquired only recently in a specific species, although this ability is currently unexplored. Nonetheless, there may be specific new evolutionary features that exceed the current reach of genomic language modeling.

The nucleotides predicted by gLMs are not only shaped by functional elements but also include mutational biases and easy-to-predict low-complexity regions that follow simple rules such as repeats. We provide preliminary evidence that analyzing nucleotide dependencies helps disentangle some of these factors, such as highly reconstructed regulatory elements compared to highly reconstructed repeats. However, development of gLM training strategies explicitly accounting for repeats[5] and mutational processes may help to further focus these models on functional elements.

So far, gLM-derived variant effect metrics leverage reconstruction probability[5,6], presuming that unlikely sequences are more deleterious. We showed that the influence of a nucleotide on predicting others is a more effective indicator of deleteriousness and could outperform alignment-based conservation. However, accounting for genetic drift and mutational biases will require research at the intersection of genomic language modeling and population genetics.

We have shown that dependency maps provide a promising new entry point to unravel the regulatory code. Regulatory elements, such as TF binding sites, manifest as dense blocks in dependency maps. We showed in yeast that applying simple image processing techniques on dependency maps identified these sites with an accuracy comparable to models trained on experimental binding data. Thus, this method is valuable for discovering regulatory elements, particularly where experimental data are limited (for example, nonmodel species, post-transcriptional regulation). Future improvements could involve modeling motifs with variable-sized blocks and accounting for all base-level dependencies. Moreover, dependencies also highlighted interactions between sequence elements in splicing and promoters, a property that future work could leverage to explore how sequence context governs the activity of regulatory elements.

Dependency maps accurately reflect bases in contacts within RNA folds, a substantial finding given the limited ground-truth data in RNA structural biology. Our entirely unsupervised approach, which relates to techniques recently proposed for unraveling amino acid contacts from protein language models[44–47], overcomes limitations of secondary structure inference, yielding information on both canonical and noncanonical contacts, pseudoknots and alternative folding. Analyzing nucleotide dependencies within RNA structure sequences is related to covariation analysis, which identifies compensating substitutions between pairs of positions in an alignment as evidence for evolutionarily conserved contacts. In contrast to covariation analysis, our approach does not require alignments, which are rarely unique and for which even a single-nucleotide shift can introduce ambiguities, affecting the covariation statistics. We note, however, that nucleotide dependency analysis and sequence-alignment-based approaches are complementary. Notably, sequence alignment often provides direct evidence of a common ancestor sequence. In contrast, gLM dependencies provide more flexibility for detecting functional interactions such as noncanonical contacts and in regimes with low alignable sequences. Furthermore, using nucleotide dependencies to infer structural contacts relies on the gLM to have been trained on enough sequences to have captured relevant evolutionary footprints. In this respect, future work could investigate the influence on the choice of species, sequences and model design.

The gLM evaluations are often based on high-level aggregate statistics, such as the area under the ROC (AUROC) curve and $R^2$, which assess the performance of downstream tasks that further models build upon. These evaluations conflate the contributions of gLMs as foundational models with those of the downstream supervised models and thus provide narrow, unidimensional assessments. Nucleotide dependencies instead enable benchmarking the gLMs themselves. We revealed critical limitations in current model architectures and single-species training practices, paving the way for more effective and generalizable gLMs.

Across various scientific fields, visualization tools also enable researchers to generate new observations and hypotheses. A nonquantifiable contribution of dependency maps, but perhaps not the least, is to allow visualizing selective constraints on sequence in a new way.

## Online content

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

## Methods

### SpeciesLM training

For SpeciesLM metazoa, we obtained metazoan genomes comprising 494 different species from the Ensembl 110 database[36]. For each annotated protein-coding gene, we extracted 2,000 bases 5′ to the start codon and trained a species-aware masked language model on this region. We followed the training and tokenization procedure outlined in Species-aware gLMs[4], but kept the batch size at 2,304, despite increasing the input sequence length, resulting in approximately twice as many tokens seen during training as in SpeciesLM fungi 5′. We used rotary positional encoding to inject positional information into the Transformer blocks.

For SpeciesLM fungi, we deviated from the above recipe by tokenizing each base of the sequences discussed in ref. 4 separately (single nucleotide, 1-mer tokenization) and using learned absolute positional encodings. To stabilize training, we increased dropout in the multilayer perceptron layers of the transformer to 0.2 and set it to 0.1 for attention dropout.

Overall, we improved the training efficiency by fusing biases of the linear layers, the multilayer perceptron in the transformer and the optimizer using Nvidia Apex. We used FlashAttention2 (ref. 51) to train all models.

### Nucleotide dependencies and variant influence score

We define the dependency between a variant nucleotide $k_{alt}$ at position $i$ and a target position $j$ as

$$e_{i,j,k_{alt}} = \max\left\{ \left| \log_2\left( \frac{\hat{\text{odds}}(n_j = k | n_1, \ldots, n_i = k_{alt}, \ldots, n_N)}{\hat{\text{odds}}(n_j = k | n_1, \ldots, n_i = k_{ref}, \ldots, n_N)} \right) \right| \right\}_{k \in \{A,C,G,T\}}$$

where $k$ is one of the four possible nucleotides A, C, G or T; $n_i$ and $n_j$ are the nucleotides at position $i$ and $j$, respectively; $k_{ref}$ is the nucleotide in the reference, nonaltered input sequence, and $k_{alt}$ is the nucleotide in the alternative sequence. The odds estimates are computed from the predictions of a gLM under consideration. For this computation, none of the nucleotides (including the target nucleotide) is masked.

The variant influence score $e_{i,k_{alt}}$, for a sequence of $N$ nucleotides, is defined by averaging the dependencies on a variant nucleotide at position $i$ across all positions $j = 1, \ldots, N$ such that $j \neq i$.

A nucleotide dependency $e_{i,j}$ between a query position $i$ and a target position $j$ on a sequence of $N$ nucleotides is given by:

$$e_{i,j} = \max\left\{ \left| \log_2\left( \frac{\hat{\text{odds}}(n_j = k | n_1, \ldots, n_i \neq k_{ref}, \ldots, n_N)}{\hat{\text{odds}}(n_j = k | n_1, \ldots, n_i = k_{ref}, \ldots, n_N)} \right) \right| \right\}_{k \in \{A,C,G,T\}}$$

We compute dependencies for all $i,j$ pairs such that $i \neq j$, that is we do not consider self-dependencies.

In autoregressive models, a query variant cannot directly affect the prediction of a target position located 5′ of the query. Thus, to obtain the lower triangular matrix of the dependency map, we also run the model on the reverse strand.

In the SpeciesLM metazoa, which predicts nucleotides as overlapping 6-mers, the procedure needs to be adapted to yield one prediction for each target nucleotide. This is achieved by first computing for each of the six 6-mers that overlap the target nucleotide of interest, which probability it implies for this target nucleotide, as previously described[4]. We then average these six probabilities to obtain a single probability.

For the Nucleotide Transformer models, which predict only nonoverlapping 6-mers, we use a similar approach. Consider the case of predicting the probability of observing nucleotide $n$ at position $i$ of the sequence. In the tokenized sequence, this nucleotide has position $p$ in the $k$th 6-mer where:

$$k = \left\lfloor \frac{i}{6} \right\rfloor$$

$$p = i \bmod 6$$

The model predicts a distribution over all $4^6$ possible 6-mers at position $k$. We first discard all predictions corresponding to 6-mers that contain a nucleotide that differs from the reference sequence at any location other than $p$—which leaves only four 6-mers. We renormalize so that the predicted probability of these remaining 6-mers sums to one. We then record the (renormalized) probability of the 6-mer that has the desired nucleotide $n$ at position $p$.

Apart from extracting nucleotide-level probabilities with the above-mentioned method, we have also experimented with computing the probability for a nucleotide at position $i$ as the sum of all k-mers containing that nucleotide at that position. Evaluation of nucleotide dependencies within tRNAs revealed a worse performance with this method.

### Variant impact benchmarks

As our metric of variant impact, we used the variant influence score. This average is computed over the full receptive field of the model for the SpeciesLM. For Nucleotide Transformer models, we only average over the central 2 kb, so as to facilitate comparisons. Nevertheless, we provide the full sequence context for which this model has been trained.

For comparison, we also calculated a variant effect score based on the gLM reconstruction at the query variant. Specifically, this score is the log ratio between the predicted probability of the variant nucleotide and the predicted probability of the reference nucleotide[5,6].

Finally, we downloaded conservation scores (PhyloP and PhastCons) for human and *S. cerevisiae* from the University of California, Santa Cruz genome browser database[21–24,52]. For humans, these include the conservation scores based on the 100-way, 447-way and 470-way alignment.

**Promoter saturation mutagenesis.** Promoter saturation mutagenesis (ref. 20) data mapped to hg38 were provided by V. Agarwal (mRNA Center of Excellence, Sanofi, Waltham, MA, USA). As discussed in ref. 29, we excluded the FOXE1 promoter due to the low replicability of the measurements, leaving nine promoters and comprising 8,635 variants. Variants were then intersected with the human gene 5′ regions (that is, the regions 2-kb 5′ of annotated start codons). Then, the variant influence score was calculated for each variant measured in the assay from the LM dependencies for these regions. The variant influence score was then correlated with the absolute value of the measured $\log_2$ fold change in expression. This correlation was computed for each promoter and then averaged across promoters.

To determine confidence intervals, we performed 100 bootstrap samples per promoter and recomputed the correlation for each bootstrap sample. The confidence interval was defined by adding/subtracting two standard deviations of the average correlation.

**eQTL variants.** For human eQTL, we downloaded SUSIE[26] fine-mapped GTEx eQTL data from EBI. We then intersected these data with the human gene 5′ regions. This procedure, by design, enriches for promoter eQTL. Similar to the details in ref. 29, we considered every eQTL variant with a posterior inclusion probability higher than 0.9 as putative causal and we considered any eQTL variant with posterior inclusion probability lower than 0.01 as putative noncausal. We only considered putative noncausal eQTL intersecting regions, which also include at least one causal eQTL. This procedure gave 2,958 eQTL variants, of which 1,631 were classified as putative causal. Then, the influence score for each variant was computed based on the nucleotide dependencies in these regions. We ranked variants according

to the influence score. Confidence intervals were computed using bootstrapping as before.

For yeast eQTL, we downloaded the results of an MPRA study assessing candidate *cis*-eQTL variants[27]. After this study, we classify any eQTL variant with false discovery rate < 0.05 in the MPRA assay as causal and we classify any eQTL with (unadjusted) *P* value of >0.2 as noncausal. This yielded 3,056 eQTL variants, of which 379 were classified as causal. These eQTL variants were then intersected with yeast gene 5′ regions and influence scores were computed from the SpeciesLM fungi dependency maps. Confidence intervals were computed using bootstrapping as before.

**Clinvar.** We used ClinVar version 2023_07_17 (ref. 19), previously downloaded from https://ftp.ncbi.nlm.nih.gov/pub/clinvar/vcf_GRCh38/. We considered noncoding any variant in the categories 'intron_variant', '5_prime_UTR_variant', 'splice_acceptor_variant', 'splice_donor_variant', '3_prime_UTR_variant', 'non_coding_transcript_variant', 'genic_upstream_transcript_variant' and 'genic_downstream_transcript_variant'. As discussed in ref. 53, we considered as pathogenic any variant classified as pathogenic or likely pathogenic and as benign any variant classified as benign or likely benign. We excluded variants with fewer than one review star. This resulted in 385,572 variants, of which 22,313 were classified as pathogenic.

As most ClinVar variants fall outside the 5′ regions of genes, we chose not to intersect with these regions. Instead, we computed the dependency map centered on the variant of interest. Confidence intervals were computed using bootstrapping as before.

**Borzoi.** We ran Borzoi in mixed precision to reduce computational overhead using the PyTorch Borzoi package. Replicate zero of Borzoi was used for all analyses. For the eQTL analysis, we computed the L2 score as discussed in ref. 25. We used the tissues of borzoi predictions matching the eQTLs. If several Borzoi tracks matched the tissue, we averaged the scores across these tracks. For ClinVar, we followed a similar approach, except that we collected Borzoi predictions for all tissues and assays. We then computed the L2 score across tracks to give a tissue-agnostic and mechanism-agnostic variant-effect score.

For the Kircher saturation mutagenesis dataset, we computed the logSED score as discussed in ref. 25. We mapped the cell types used in the assay to Borzoi tracks as follows: for the GP1BB, HBB, NHBG1 and PKLR promoters, we used 'RNA:K562'; for the F9 and LDLR promoters, we used 'RNA:HepG2'; for HNF4A and MSMB, we used 'RNA:kidney' (as HEK293 is originally a kidney cell) and for TERT, we used 'RNA:astrocyte' (as glioblastoma are cancerous astrocytes).

**Integrative model using Borzoi and the influence score**
We integrated Borzoi and the influence score using logistic regression—for the eQTL and ClinVar predicitions—and using linear regression for the mutagenesis data using fivefold cross-validation scheme for all benchmarks. Notably, for the Kircher saturation mutagenesis task, model fitting and cross-validation were performed separately for each promoter, and performance was then averaged across folds and promoters.

**Alternative dependency metrics**
All benchmarks on alternative dependency metrics were performed on the SpeciesLM fungi.

**Gradient-based.** We computed the gradient of the prediction for each nucleotide at position *i* with respect to each nucleotide at position *j* yielding a 4 × 4 matrix. To achieve this, we first replaced the tokenization layer with a one-hot encoding and a linear layer, which map the one-hot encoded nucleotides to their respective token embeddings. We then propagated gradients from each target nucleotide prediction

to each one-hot encoded input nucleotide. As a metric of nucleotide dependency, we then used the maximum absolute value across the 4 × 4 matrix of each *i,j* position.

**Mask-based.** Masked-based dependencies are computed as:

$$e_{i,j} = \max\left\{\left|\left|\log_2\left(\frac{\text{odds}\left(n_j = k | n_1, \dots, n_i = [\text{MASK}], \dots, n_N\right)}{\text{odds}\left(n_j = k | n_1, \dots, n_i = k_{\text{ref}}, \dots, n_N\right)}\right)\right|\right|\right\}_{k \in \{A,C,G,T\}}$$

where '[MASK]' stands for the mask token, *k* belongs to one of the four possible nucleotides A, C, G or T; $n_i$ and $n_j$ are the nucleotides at position *i* and *j*, respectively; $k_{\text{ref}}$ is the nucleotide in the reference, nonaltered input sequence.

### *S. cerevisiae* tRNA structure benchmark
*S. cerevisiae* genome assembly version R64-1-1 and annotation version R64-1-1.53 were downloaded from EnsemblFungi[36]. The *S. cerevisiae* tRNA secondary structures were downloaded from GtRNAdb[54]. We considered only the tRNAs overlapping the 1 kb 5′ regions to any yeast start codon, yielding 172 tRNA sequences. Subsequently, dependency maps on tRNAs were processed by taking the maximum between $e_{i,j}$ and $e_{j,i}$. This symmetrizes the dependency map and achieves one unique score per pair of positions in the tRNA sequence. We then used this score to predict whether a pair of nucleotides belonged to a secondary structure contact.

### Assessment of donor–acceptor dependencies in *S. cerevisiae*
We extracted intron sequences by selecting the regions within annotated gene intervals that lie between exon annotations. This resulted in 380 sequences. We then retained only introns bounded by canonical splice site dinucleotides GT and AG, yielding 272 sequences. We then computed the average dependency between every donor and acceptor nucleotide within the intron as a measure of dependency between the donor and acceptor sites. We designed two negative sets for a given intron. For the negative set 'Decoy acceptor', we compute the average dependency between donor nucleotides and each AG dinucleotide within the intron that does not include the acceptor site. For the negative set 'Matched distance', we sampled four random dependencies between nucleotides that were as distant from each other as the donor was from the acceptor, without including the donor or acceptor themselves.

### TF motif mapping
We downloaded FIMO PWM scan results from http://www.yeastss.org (ref. 55) and Chip-Exo TF binding peaks from http://www.yeastepigenome.org (ref. 31). We then extracted all Chip-Exo peaks for the available PWMs. We excluded PWM matches for which no Chip-Exo data were available for the corresponding factor. This procedure yielded data for 68 TFs. We annotated every nucleotide within 1 kb 5′ of a start codon as part of a binding TF motif if it is (1) part of a PWM match with *P* value of <0.01 and (2) this PWM match is within ten bases of a Chip-Exo peak of the corresponding TF. We defined the positive class in this way to ensure that we capture nucleotides relevant for determining binding (that is, motif) rather than all nucleotides close to a Chip-Exo peak, regardless of their role in binding. This resulted in 92,117 binding nucleotides out of a total of 6,538,427. We designated a nucleotide as repeat if it was masked by RepeatMasker. We extracted this information from the soft-masked GTF provided by Ensembl[36].

### The 69-way alignment
We used progressive cactus[56] to align 69 budding yeast species using default parameters and specifying *S. cerevisiae* as reference quality genome. We then extracted fourfold degenerate sites and used phyloFit with the EM algorithm to estimate a neutral model. Using this neutral

model and the alignment, we ran phastCons with the parameters --rho 0.3 --estimate-rho --target-coverage 0.4 --expected-length 23, which correspond to the parameters used in the seven-way alignment[21]. We also ran phyloP[22], with --method LRT --mode CONACC.

### Dependencies in rare-variant-associated aberrant splicing

We computed dependency maps for all rare SNVs associated with splicing outliers in GTEx[28] as described earlier[32]. Because the input length of SpliceBert[14] is limited to 1,024 bp, the complete set of variant outlier pairs ($n = 18,371$) was filtered such that the variant and associated outlier junction were located within an 800-bp window ($n = 1,811$) and 100 bp of sequence was added from the maximum and minimum positions of the variant and outlier junction splice sites. For each variant location, we extracted the average value of the dependency map at the intersection of either variant and outlier donor dinucleotide or variant and outlier acceptor dinucleotide. This variant effect score was compared against a background score. This background score was computed as the mean over all dependencies that were as distant from each other as the variant was from the outlier donor (matched distance) or the outlier acceptor. The scores were filtered for a minimum distance of 5 bp between the variant and splicing dinucleotide to filter values near the diagonal corresponding to self-interactions. Variant categories were annotated with the Ensembl variant effect predictor (VEP)[57]. For each variant, the most severe VEP annotation was considered. For the 'exon' category, the following VEP categories were grouped together: synonymous_variant, missense_variant, stop_lost, stop_gained.

### Genome-wide search for parallel and antiparallel dependencies

We scanned dependency maps for parallel and antiparallel dependencies using 5 × 5 convolutional filters. We constructed the antiparallel filter by populating the antidiagonal of a zero-filled 5 × 5 matrix with ones, and for the parallel filter, by populating the diagonal with ones. We then centered each filter by subtracting the mean value from each position to ensure that a convolution on a uniform 5 × 5 region yields a result of zero. We applied these filters to dependency maps from SpeciesLM fungi (both filters) and RiNALMo (antiparallel filter only)[4,7].

### Search for parallel and antiparallel dependencies in fungi using the SpeciesLM fungi

For the SpeciesLM fungi, we have computed dependency maps spanning 1 kb 5′ of each annotated start codon on a set of representative fungi species, including *Agaricus bisporus*, *Candida albicans*, *Debaryomyces hansenii*, *Kluyveromyces lactis*, *Neurospora crassa*, *S. cerevisiae*, *S. pombe* and *Yarrowia lipolytica*. The genomes and annotation files for each species were downloaded from EnsemblFungi release 53 with accessions GCA_000300555.1, GCA000182965v3, GCA_000006445.2, GCA000002515.1, GCA_000182925.2, GCA_003046715.1, GCA_000002945.2 and GCA_000002525.1, respectively.

All regions annotated as 'five_prime_utr', 'three_prime_utr', 'intron', 'CDS', 'pseudogene_with_CDS' and other regions (for example, nonannotated introns) inside an annotated gene interval were categorized as protein-coding gene. All regions annotated as 'tRNA', 'tRNA_pseudogene', 'rRNA', 'snRNA', 'ribozyme', 'SRP_RNA', 'snoRNA', 'RNase_P_RNA' and 'RNase_MRP_RNA' were categorized as structured RNA. Finally, all regions annotated as 'transposable_element', 'pseudogene' and regions without any annotation were considered as intergenic.

### Search for antiparallel dependencies and RNA structure in *E. coli* using RiNALMo

For RiNALMo, we computed dependency maps for regions 100, 200 and 500 bp before each annotated start codon in *E. coli* str. K-12 substr. MG1655, whose genome and annotation were downloaded from GenBank[58] with accession U00096.3.

As candidates for a new RNA structure, we first filtered positions whose convolution value is greater or equal to 25 to select only high-value antiparallel dependencies, resulting in a filtered convolved dependency map. Next, we counted the unique number of antidiagonals potentially belonging to one stem by extracting the unique $i + j$ nonzero positions supported by at least three nonzero values.

As candidates for a new structure, we selected maps suggesting the existence of at least two potential stems.

### RNA secondary structure benchmarking

We downloaded the database of secondary structures Archive II[37], which includes 3,865 curated RNA structures across nine families (5S rRNA, SRP RNA, tRNA, tmRNA, RNase P RNA, group I intron, 16S rRNA, telomerase RNA and 23S rRNA). For each structure, we generated the dependency map with the pretrained RiNALMo and retained the largest of the two dependency map entries for each pair of nucleotides (maximum of $i,j$ and $j,i$). The AUROC curve was computed for each structure against the Archive II secondary structure annotations.

### Benchmarking of canonical and noncanonical RNA contacts

We downloaded the database of RNA structures CompaRNA[38], which is a compilation of RNA contacts based on 201 available RNA structures in the Protein Data Bank by RNAView[59]. Contacts are classified either as 'standard' or as 'extended'. While the first includes only canonical AU, GC and wobble GU pairs in the *cis*-Watson–Crick/Watson–Crick conformation[60], the latter calls all interacting bases regardless of their conformation, including noncanonical or tertiary contacts. Of the 201 structures, 196 had a length below the maximum input length of RiNALMo (1,022 nt). For each structure, we generated the dependency map using the pretrained RiNALMo and retained the largest entry from the two dependency maps for each pair of nucleotides. Similarly, the same structures were also evaluated with the fine-tuned RiNALMo model version rinalmo_giga_ss_bprna_ft, resulting in a predicted value for each pair of nucleotides. To evaluate their performance in predicting noncanonical contacts, we excluded all canonical contacts and computed the AUROC curve for all remaining positions across all structures. Significance between ROC AUCs was determined by bootstrapping over 10,000 permutations.

### Comparison with RNAalifold

We evaluated the performance of the dependency maps against RNAalifold[61], a standard alignment-based method for predicting a consensus RNA structure by incorporating sequence covariation from a set of aligned RNA sequences as input. For this, we use the 201 PDB entries in CompaRNA that had at least one Rfam match and consider two subsets. The first subset consisted of the 33 PDB sequences that contained an exact sequence match between the PDB entry and at least one Rfam seed alignment. In case of multiple matching Rfam seed alignments (for example, ribosomal RNA), we considered an arbitrarily chosen single Rfam seed alignment to avoid confounding the evaluations by duplicates. The second subset consisted of the remaining 168 sequences. For this, we used nhmmer (v3.1b2)[62] to find homologous sequences within a database of 220,478 bacterial and archaeal genomes and plasmids downloaded from NCBI. After removing sequences longer than 1,022 nt (the maximum context length for which the gLM RiNALMo has been trained), this resulted in 67 sequences with hits in the database.

On the first subset, we use the Rfam seed alignments as input to RNAalifold. To assess the robustness of the analyses to the alignment procedure, we additionally realigned the sequences in the seed alignments using Clustal-Omega (v1.2.4)[63] and MAFFT (v7.525)[64]. For the second subset, we performed sequence alignments using both Clustal-Omega and MAFFT, limiting the alignments to a maximum of 1,000 sequences (by aligning the PDB sequence to the top 999 nhmmer hits) to reduce computation time. On both subsets and from each

alignment, a base-pair probability matrix corresponding to the predicted RNA structure was generated using RNAalifold available through the ViennaRNA (v2.6.4) package[65]. RNAalifold was run in the following two modes: using the default energy model (command: RNAalifold -p) and with RIBOSUM scoring (command: RNAalifold -p -r).

## Pseudoknot benchmark

We downloaded the compendium dataset bpRNA-1m(90) that contains 28,370 annotated RNA structures with less than 90% sequence similarity obtained from the databases CRW, tmRNA, SRP, tRNAdb2009, RNP, RFAM and PDB[39]. From these, we extracted all structures that contain pseudoknot contacts and are no longer than 1,022 nt (the maximum context length for which the gLM RiNALMo has been trained). These resulted in 2,530 structures of varying lengths and sources. We then extracted the pseudoknot contacts from the dot-bracket notation provided by bpRNA that takes into account non-nested pairs[39]. Finally, we computed the dependency maps for each one of these structures and evaluated their ability to predict whether a pair of nucleotides belongs to a pseudoknot contact (positive set) or does not belong to a structure contact (pseudoknot or canonical structure contact—negative set).

## DMS-MaPseq analysis of *E. coli* cells

*E. coli* TOP10 cells were grown in LB broth at 37 °C with shaking until $OD_{600} = 0.5$, after which dimethyl sulfate (DMS; Sigma-Aldrich, D186309), prediluted 1:4 in ethanol, was added to a final concentration of 200 mM. Bacteria were incubated for 2 min at 37 °C, and reaction was quenched by addition of 0.5 M final DTT. Bacteria were pelleted by centrifugation at 17,000$g$ for 1 min at 4 °C, after which they were resuspended in cell pellets in 12.5-μl resuspension buffer (20 mM Tris–HCl pH 8.0; 80 mM NaCl; 10 mM EDTA pH 8.0), supplemented with 100 μg ml⁻¹ final lysozyme (L6876, Merck) and 20 U SUPERase·In RNase Inhibitor (Thermo Fisher Scientific, A2696), by vortexing. After 1 min, 12.5-μl lysis buffer (0.5% Tween-20; 0.4% sodium deoxycholate; 2 M NaCl; 10 mM EDTA) were added, and samples were incubated at room temperature for 2 additional min. Then 1 ml TRIzol Reagent (Thermo Fisher Scientific, 15596018) was added, and RNA extracted as per the manufacturer's instructions. rRNA depletion was performed on 1 μg total RNA using the RiboCop for Bacteria kit (Lexogen, 126). DMS-MaPseq library preparation was performed as previously described[41]. After sequencing, reads were aligned to the *E. coli* str. K-12 substr. MG1655 genome (GenBank, U00096.3), using the rf-map module of the RNA framework[66] and Bowtie2 (ref. [67]). Count of DMS-induced mutations and coverage and reactivity normalization were performed using the rf-count-genome and rf-norm modules of the RNA framework. Experimentally informed structure modeling was performed using the rf-fold module of the RNA framework and ViennaRNA (v2.5.1)[67].

## RNA structure covariation analysis

Covariation analysis was performed using the cm-builder pipeline (https://github.com/dincarnato/labtools) and a nonredundant database of 7,598 representative archaeal and bacterial genomes (and associated plasmids, when present) from RefSeq[68].

## Evaluation of artificial forward and inverted duplications

We generated random sequences of 100 nucleotides by sampling from regions 1 kb 5′ of the start codon in *S. cerevisiae* to ensure a representative GC content and shuffling the sequences to destroy potential functional elements. Additionally, we created 100 unique duplicated sequences, ranging from 2 to 20 nucleotides in length, by randomly sampling each nucleotide with equal probability. Each duplicated sequence was then inserted into a uniquely generated 100-nucleotide sequence at a random distance from each other, ensuring no overlaps occurred. We used the SpeciesLM fungi to generate dependency maps for each sequence. We then computed average dependencies by taking the mean of the dependencies between

nucleotides and their duplicates. This involved averaging across a parallel diagonal for forward duplications and an antiparallel diagonal for inverted duplications.

For tRNA-sized sequences, we followed a similar method but generated each sequence by shuffling each unique tRNA sequence in *S. cerevisiae* once. We computed the average number of inverted duplications by averaging the occurrences of duplicated sequences of specific lengths across 10,000 shuffled versions of each tRNA sequence.

## Genome-wide analysis of dependency distribution

Using the SpeciesLM fungi, we computed dependency maps across the genomes of *S. cerevisiae* and *S. pombe*. Because the SpeciesLM fungi was pretrained on sequences of 1,003 nucleotides, including the start codon at the end, we discarded dependencies involving the last three nucleotides of each sequence, yielding dependencies for 1,000 nucleotides. Genome-wide dependency maps of 1-kb span were obtained with a tiling approach. Along each chromosome, we computed 1-kb square dependency maps every 500 bp and averaged overlapping entries.

To ensure that the same number of targets is computed before and after a specific query nucleotide, we considered dependencies involving nucleotides at most 500 positions away from each other. For each map, we sampled 1,000 dependencies and averaged dependencies mapping to the same genomic positions but computed from different overlapping maps. Due to limitations in numerical precision, we considered only dependencies larger than 0.001.

To compute the power–law coefficients, a linear regression was fitted to predict the logarithm of the dependency from the logarithm of its corresponding distance in nucleotides. The scaling coefficient was then obtained by exponentiating the fitted intercept of the linear regression, and the decay rate was obtained directly from the fitted slope. The scaling coefficient and decay rate were computed for different regions in the genome which are as follows: (1) nuclear—involving all dependencies belonging to nuclear DNA; (2) mitochondria—involving all dependencies within mitochondrial DNA; (3) structured RNA—belonging to the annotations 'tRNA', 'tRNA_pseudogene', 'rRNA', 'snRNA', 'ribozyme', 'SRP_RNA', 'snoRNA', 'RNase_P_RNA' or 'RNase_MRP_RNA'; (4) protein-coding gene—belonging to the annotations 'five_prime_utr', 'three_prime_utr', 'CDS' or 'pseudogene_with_CDS'; (5) intron—belonging to the regions inside an annotated gene interval but not to exons and (6) intergenic—belonging to all regions annotated as 'transposable_element', 'pseudogene', as well as regions without any annotation.

## Model comparison

All other models used were downloaded from Huggingface or from their publicly available repositories. Human tRNA sequences were downloaded from GtRNAdb[54]. Exact duplicate sequences were removed, leaving 266 tRNAs.

## Reporting summary

Further information on research design is available in the Nature Portfolio Reporting Summary linked to this article.

## Data availability

Data to reproduce the analysis, together with the 69-Saccharomycetales genome alignment and conservation score, as well as the bacterial and archaeal genomes and the plasmid sequences used for the benchmark against RNAalifold, are provided in ref. [69]. The SpeciesLM models are available at https://huggingface.co/collections/johahi/specieslms-678a39261cfff01c1fa3ae41. Raw DMS-MaPseq data have been deposited to the Gene Expression Omnibus database under accession GSE271937.

## Code availability

The code required to reproduce the results in the paper is available at https://github.com/gagneurlab/dependencies_DNALM or in ref. [69].

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

## Acknowledgements

P.T.d.S. is supported by the Munich Center for Machine Learning. N.W. is supported by the Helmholtz Association under the joint research school 'Munich School for Data Science—MUDS'. D.I. was supported by the Dutch Research Council (NWO), NWO Open Competitie ENW—XS (project OCENW.XS22.1.015) and by the European Research Council (ERC), European Union's Horizon Europe research and innovation program (grant agreements 101124787 and RNAStrEnD). X.H.-A. was supported by an EMBO Postdoctoral Fellowship (ALTF 792-2022). J.G. was supported by the German Bundesministerium für Bildung und Forschung (BMBF) through the Model Exchange for Regulatory Genomics project MERGE (031L0174A). J.G. was also supported by the Deutsche Forschungsgemeinschaft (DFG; German Research Foundation) through the project NFDI 1/1 'GHGA—German Human Genome-Phenome Archive' (441914366), and funded through the EVUK program ('Next-generation AI for Integrated Diagnostics') of the Free State of Bavaria. J.G. and G.S.T.G. are supported by the DFG (German Research Foundation) through the TRR267 (403584255). This study was supported by the DFG (German Research Foundation) through the IT Infrastructure for Computational Molecular Medicine (project 461264291). This study was also supported by the ERC (EPIC; 101118521 to P.T.d.S., A.K., J.H., N.W. and J.G.) and by the European Union. Views and opinions expressed are, however, those of the author(s) only and do not necessarily reflect those of the European Union or the ERC Executive Agency. Neither the European Union nor the granting authority can be held responsible for them. We thank F. Bonneau for assistance with the presentation of the three-dimensional structure of a tRNA and further thank S. Aerts and J. Cheng for the useful feedback on the manuscript.

## Author contributions

P.T.d.S. and A.K. conceptualized the study and performed the methodology, software development, formal analysis, investigation and visualization, and contributed to writing the original draft and writing, reviewing and editing the final draft of the paper. P.T.d.S. managed project administration. J.H.performed the methodology, software development and investigation, and contributed to writing, reviewing and editing the final draft of the manuscript. G.S.T.G., N.W. and X.H.-A. performed the methodology, software development, investigation and formal analysis, and contributed to writing, reviewing and editing the final draft of the paper. D.I. conducted formal analysis, investigation, resource management and data curation; contributed to writing, reviewing and editing the final draft of the paper, and was responsible for visualization, supervision and funding acquisition. J.G. conceptualized the study and performed methodology and resource management; contributed to writing the original draft and writing, reviewing and editing the final draft of the paper; and was responsible for visualization, supervision, project administration and funding acquisition.

## Funding

## Competing interests

The authors declare no competing interests.

## Additional information

**Extended data** is available for this paper at https://doi.org/10.1038/s41588-025-02347-3.

**Correspondence and requests for materials** should be addressed to Julien Gagneur.

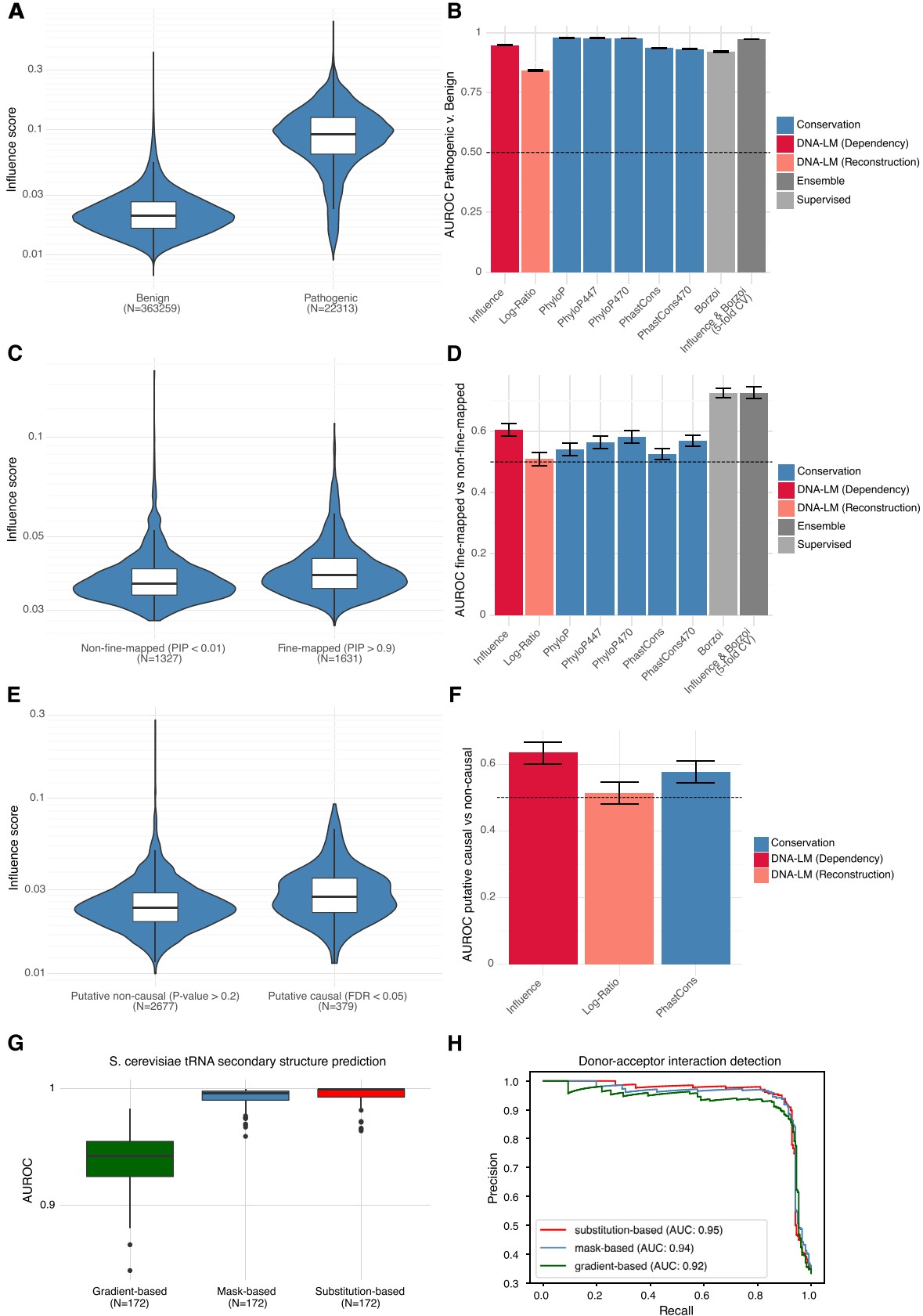

**Extended Data Fig. 1 | See next page for caption.**

**Extended Data Fig. 1 | Nucleotide dependencies capture functional interactions and variant effects in ClinVar and eQTLs. a**, Variant influence score against pathogenic and benign variants classified from ClinVar. P-value obtained from double-sided Wilcoxon rank-sum test <10⁻⁶. **b**, Performance (AUROC) for the classification of ClinVar variants into pathogenic or benign comparing the variant influence score, gLM log ratio between predicted probability of the reference nucleotide and variant nucleotide, alignment-based conservation scores from PhyloP and PhastCons, the supervised model Borzoi, as well as a logistic regression on the influence-score and Borzoi (the latter was fitted and evaluated using 5-fold cross-validation on this dataset). Pathogenic N = 22313; benign N = 363259. **c**, Variant influence score for putative causal and putative non-causal variants as obtained from fine-mapped human eQTL[26,70]. P-value obtained from double-sided Wilcoxon rank-sum test <10⁻⁶. **d**, Area under the receiver operating characteristic curve (AUROC) for the classification of putative causal versus putative non-causal variants from the fine-mapped human eQTL.

Ensemble model fitted as in **b**. Non-fine-mapped N = 1327; fine-mapped N = 1631. **e**, Variant influence score for putative causal and putative non-causal variants obtained from yeast eQTL[27]. P-value obtained from double-sided Wilcoxon rank-sum test <10⁻⁶. **f**, Performance in AUROC for the classification of yeast putative causal vs putative non-causal eQTL variants. Putative non-causal N = 2677; putative causal N = 379. **g**, Performance (AUROC) for the prediction of tRNA secondary structure contacts using different nucleotide dependency metrics: gradient-based, mask-based and substitution-based. **h**, Precision-recall curves for the prediction of splice site interactions using different nucleotide dependency metrics as before. Donor-acceptor interactions N = 238; non-donor-acceptor interactions at matched distances N = 476. For all boxplots: center line, median; box limits, first and third quartiles; whiskers span all data within 1.5× interquartile ranges of the lower and upper quartiles. All error bars represent ±2 standard deviations, constructed using 100 bootstrap samples. The height of each bar corresponds to the AUROC using the different variant scores.

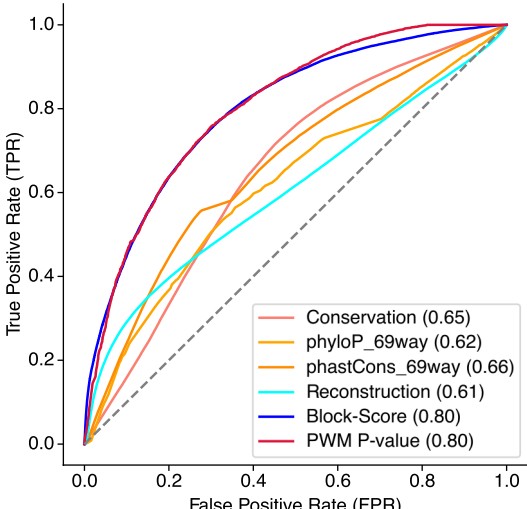

**Extended Data Fig. 2 | The block-score also outperforms other metrics in identifying transcription factor (TF) binding sites when restricted to non-coding regions.** Receiver operating characteristic (ROC) curve comparing the ability of different metrics to classify whether a nucleotide is part of a bound TF motif or not (92,117 binding nucleotides out of 3,334,202 overall). As in Fig. 2d, but overlap with coding sequences was removed. This improves the performance of alignment-based conservation somewhat, but the block-score still performs much better. Computing conservation scores on a 69-way alignment of budding yeast species did not improve discrimination.

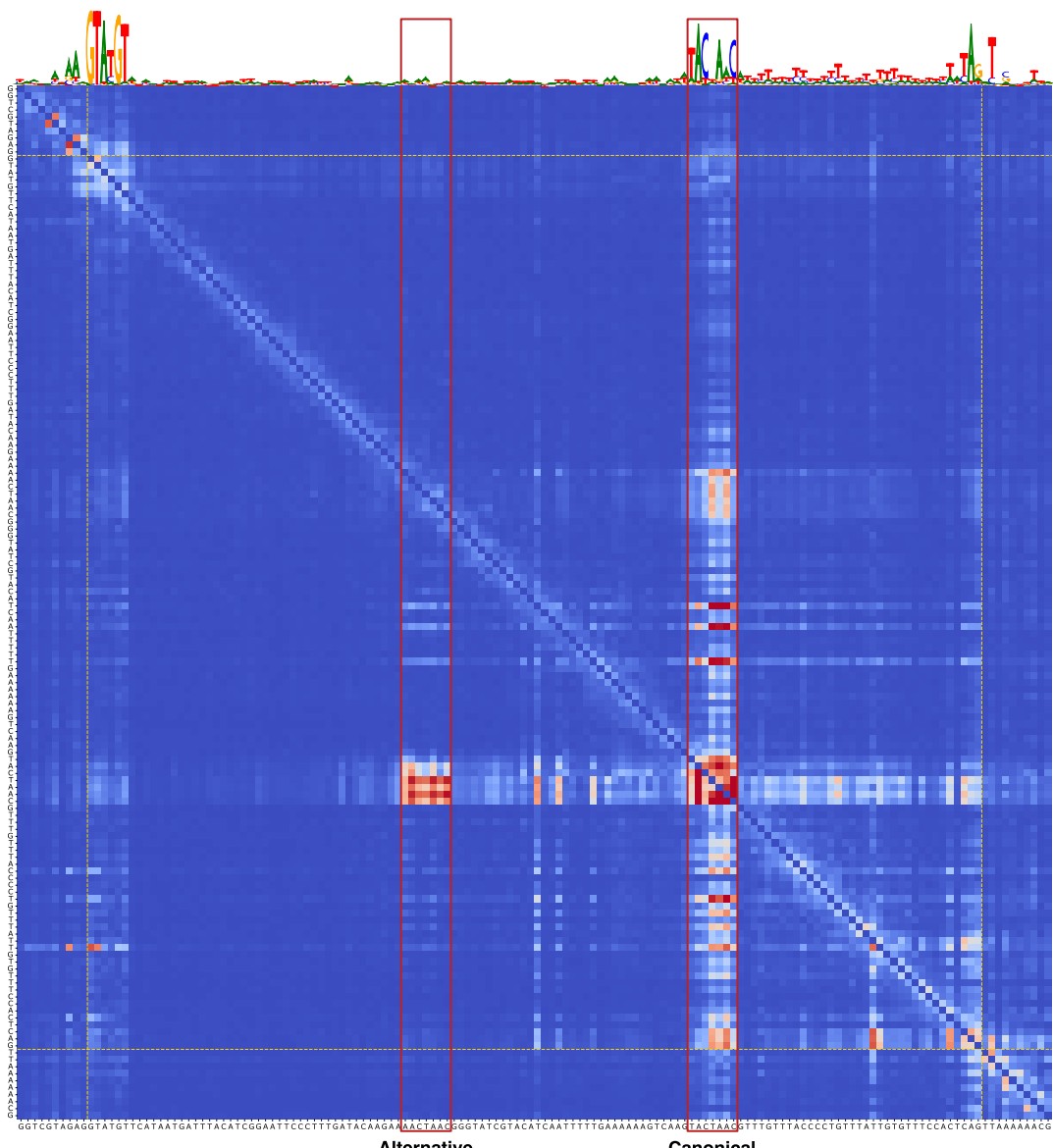

**Extended Data Fig. 3 | Dependency map highlights an alternative branch-point of the yeast gene _LSM2_.** Dependency map for an intron of the yeast gene _LSM2_ not only highlighting the canonical donor, acceptor and branch point but also an alternative non-canonical branch point. While the canonical branch point appears as an on-diagonal block, another parallel off-diagonal block is visible, suggesting that if mutations altered the canonical branch point then compensatory mutations on the alternative branch points would be favored. The target nucleotides of this block belong to a branch-point-like sequence, indicating a role as an alternative branch-point, which has also been previously found experimentally[71].

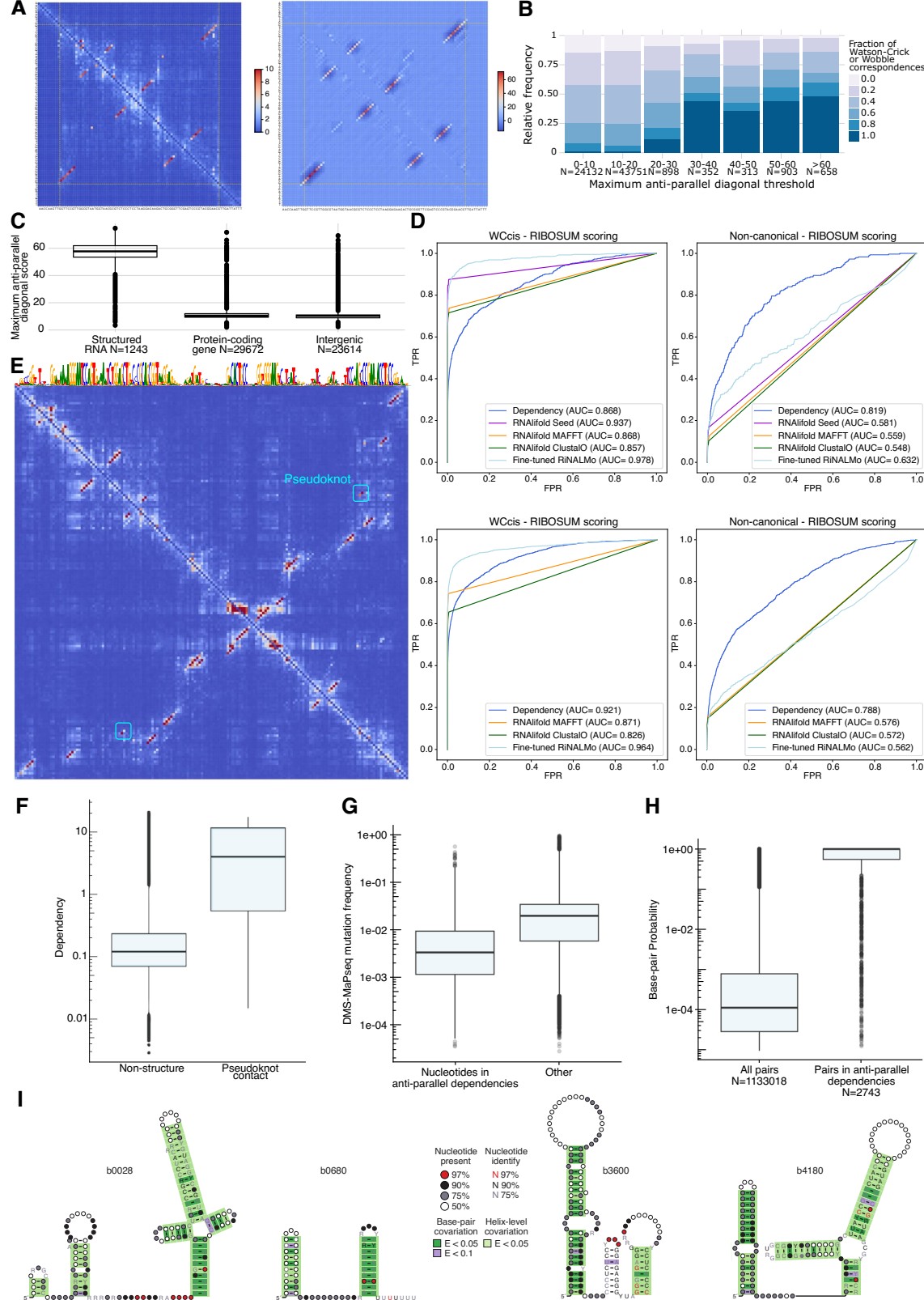

**Extended Data Fig. 4 | See next page for caption.**

**Extended Data Fig. 4 | Nucleotide dependencies systematically highlight RNA secondary structure, pseudoknot and non-Watson-Crick contacts.**
**a**, Example of a convolution of a 5 × 5 anti-parallel diagonal filter on the dependency map of yeast tR(ACG)O tRNA. The dependency map is shown on the left, while the resulting convolution is shown on the right. The maximum convolution values highlight the anti-parallel dependencies within the tRNA. **b**, Fraction of nucleotides that show Watson-Crick or wobble correspondence within the 5 base-pair region defined by the convolution filter location with the maximum hit for different convolution values in a region of 1 kb 5′ of a start codon. **c**, Maximum anti-parallel dependencies within a 1-kb region 5′ of each start codon across fungi species and its location in the genome categorized in one of structured RNA, protein coding (spanning the whole interval of a protein-coding gene), and intergenic (spanning mostly non-annotated regions between genes). **d**, ROC curve for classifying experimentally obtained canonical (left) and non-canonical (right) contacts from the compaRNA database. The performance of RNAalifold base-pair probabilities, the fine-tuned RiNALMo and nucleotide dependencies is shown for two regimes: (1) high-quality manually curated seed alignments from the Rfam database together with realignment of the same sequences using MAFT and ClustlO (top) (2) alignments constructed from a database search of 220,478 bacterial and archaeal genomes and plasmids downloaded from NCBI (bottom). **e**, *E. coli* cobalamin riboswitch dependency map together with the highlighted pseudoknot contacts and nucleotide reconstruction on top. **f**, Distribution of dependencies for pairs of nucleotides belonging to an annotated pseudoknot contact (right, N = 21,051) or not belonging to a structure contact (left, N = 175,016,129). Dependencies discriminate between these two categories (area under the ROC curve 0.92, double-sided Wilcoxon Rank-sum test $P < 10^{-16}$). All dependencies were computed for RNAs with pseudoknot contacts across 2,530 structures in bpRNA spanning multiple species and database sources (Methods). **g**, Distribution of DMS mutation frequencies for all A and C nucleotides (the nucleotides probed by DMS) in *E. coli* non-coding regions in antiparallel dependencies against all remaining non-coding nucleotides. Nucleotides part of dependency-map antiparallel stretches have significantly lower mutation rates ($P < 10^{-16}$, double-sided Wilcoxon rank-sum test, nucleotides in anti-parallel dependencies N = 5,008, other N = 103,867), which indicates they were more protected from DMS and therefore more likely to be in Watson-Crick base pairing. **h**, Ground-truth base-pairing probabilities derived from DMS-MaPseq experimentally-constrained RNA secondary structure prediction for nucleotide pairs in anti-parallel dependencies against all pairs ($P < 10^{-16}$, double-sided Wilcoxon rank-sum test, all pairs N = 1,133,018, pairs in anti-parallel dependencies N = 2,743). **i**, Covariation analysis of four novel structures validated by DMS-MaPseq 5′ of genes FkpB (b0028), glnS (b0680), mtlD (b3600) and rlmB (b4180). For all boxplots: center line, median; box limits, first and third quartiles; whiskers span all data within 1.5× interquartile ranges of the lower and upper quartiles. P values were computed using the paired two-sided Wilcoxon test.

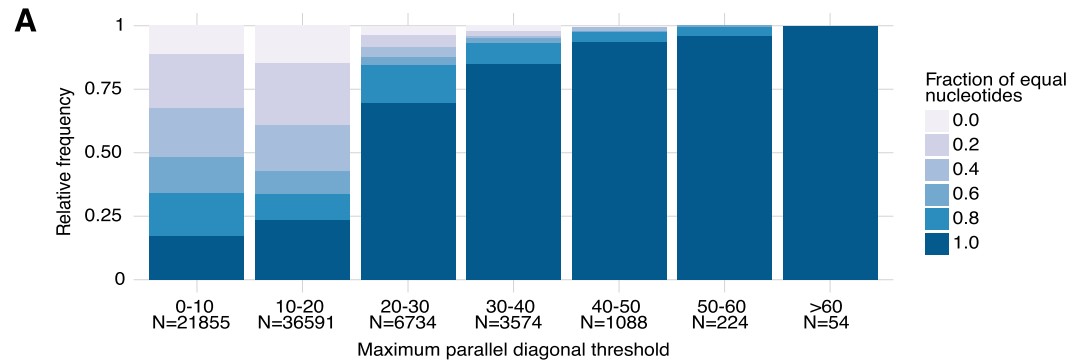

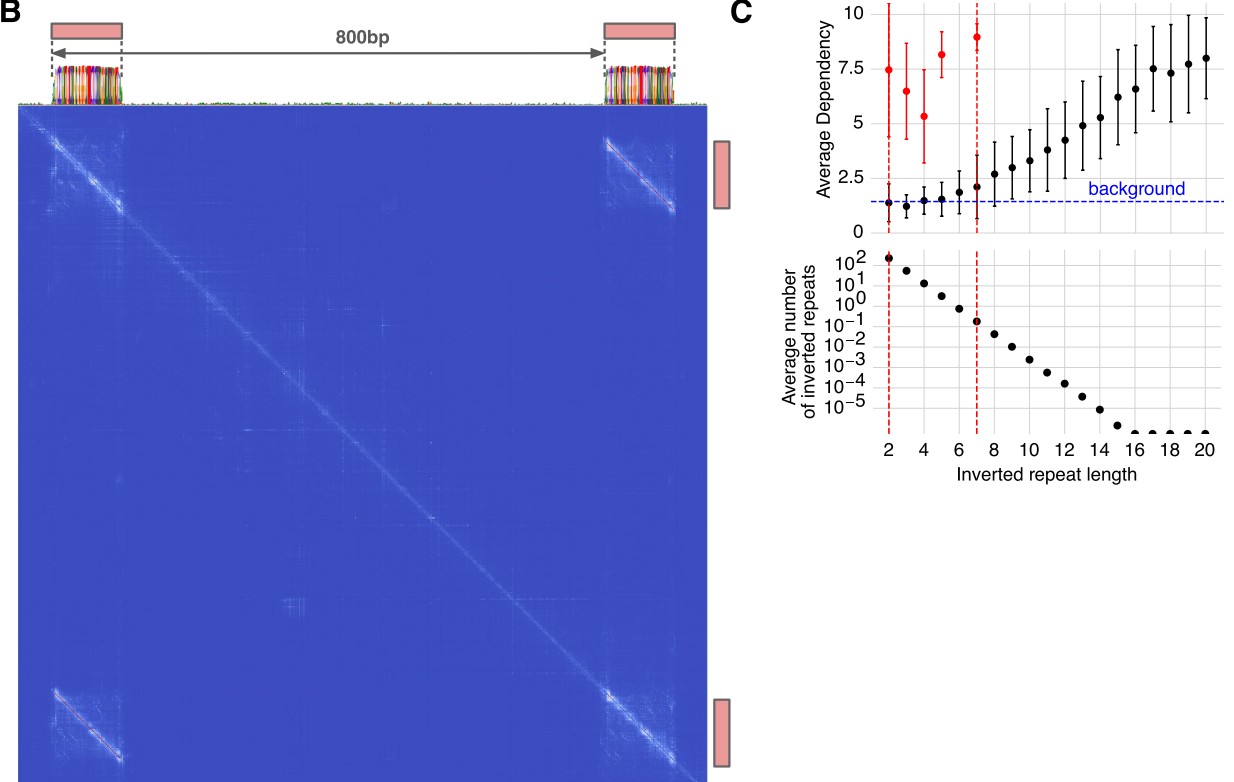

**Extended Data Fig. 5 | gLMs capture repeated sequences genome-wide but distinguish between inverted repeats within and outside structural contacts. a**, Fraction of equal nucleotides within the 5 base-pair region defined by the parallel diagonal convolution filter location with the maximum hit for different convolution values. The strongest parallel diagonal dependencies belong to repeated sequences, indicating that repeats are highlighted genome-wide in parallel dependency patterns. **b**, Dependency map and nucleotide reconstruction for a 1 kb random sequence containing an inserted artificially generated random duplicated sequence of 100b. Despite the repeated nucleotides being spaced 800 bp apart, the gLM highlights the parallel dependency linking each nucleotide. **c**, Top, average dependency against inverted repeat length for tRNA length sequences (black colored dots). The red colored dots indicate the average dependency within anti-parallel dependencies in tRNA stems. Error bars indicate 95% confidence intervals across 100 simulated sequences (black) or all tRNAs with specific stem lengths (red). Bottom, average number of inverted repeats expected to get by chance for each repeat length.

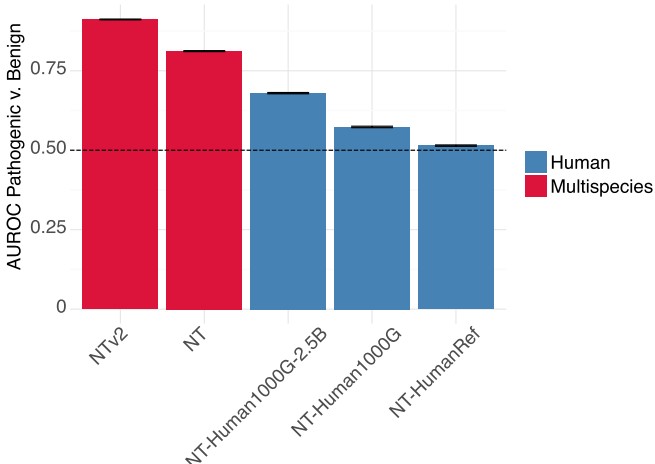

**Extended Data Fig. 6 | Performance of the nucleotide transformer models on ClinVar variant pathogenicity prediction.** Area under the ROC curve for absolute variant effect prediction on the same dataset as Extended Data Fig. 1a using variant influence scores computed from Nucleotide Transformer models.

Error bars represent ±2 standard deviations, constructed using 100 bootstrap samples. The height of each bar corresponds to the AUROC using the different model scores.

**Extended Data Table 1 | All gLMs assessed in this study together with their input sequence specifications, training data, architecture, and figure panels where they are used**

| Model | Context | Resolution | Training Data | Architecture | Specialized for | Used in |
|---|---|---|---|---|---|---|
| SpeciesLM Fungi | 1 kb | Single bp | Fungi regions 5' of start codons (806 species) | Transformer | Promoters, 5'UTR, quasi Genome-wide* | Fig. 1D, Ext. Fig. 1D,E,G,H, Fig. 2A,C,D,E, Fig. 3B,C, Ext. Fig. 3, Ext. Fig. 4A,B,C, Fig. 5A,B, Ext. Fig. 5A,B,C, Fig. 6A,B,C,D, Fig. 7A,B |
| SpeciesLM Metazoa | 2 kb | Single bp (overlapping 6-mer) | Metazoa regions 5' of start codons (494 species) | Transformer | Promoters, 5'UTR | Fig 1C, Ext. Fig. 1A,B,C,F, Fig. 2B, Fig.3A |
| SpliceBERT | 512 bp | Single bp | Metazoa pre-mRNA | Transformer | Splicing | Fig. 3D,E |
| RiNALMo | 1 kb | Single bp | noncoding RNA | Transformer | noncoding RNA | Fig. 4A,B,C,D,E,F,G, Fig. 7A,B |
| Nucleotide Transformer NTv2 | 12 kb | 6-mer | Multispecies Genomes (Metazoa, Fungi, Prokaryotes) | Transformer | Genome-wide | Fig 7A,B,C, Ext. Fig. 7A |
| Nucleotide Transformer NT | 6 kb | 6-mer | Multispecies Genomes (Metazoa, Fungi, Prokaryotes) | Transformer | Genome-wide | Fig. 7B,C, Ext. Fig. 7A |
| Nucleotide Transformer NT–Human1000G-2.5B | 6 kb | 6-mer | 3,202 diverse Human genomes | Transformer | Genome-wide | Fig. 7A,B,C, Ext. Fig. 7A |
| Nucleotide Transformer NT–Human1000G | 6 kb | 6-mer | 3,202 diverse Human genomes | Transformer | Genome-wide | Fig. 7B,C, Ext. Fig. 7A |
| Nucleotide Transformer NT–HumanRef | 6 kb | 6-mer | Human Reference Genome | Transformer | Genome-wide | Fig. 7A,B,C, Ext. Fig. 7A |
| DNA-BERT | 512 b | Single bp (overlapping 6-mer) | Human Reference Genome | Transformer | Genome-wide | Fig. 7B |
| Hyena | 1 Mb** | Single bp | Human Reference Genome | Hyena (Autoregressive) | Genome-wide | Fig. 7A,B |
| Evo | 8 kb** | Single bp | Prokaryotic Genomes | Striped Hyena (Autoregressive) | Genome-wide | Fig. 7A,B |
| CaduceusPS | 131 kb | Single bp | Human Reference Genome | Bidirectional Mamba | Genome-wide | Fig. 7A,B |
| PlantCaduceus | 512 b | Single bp | 16 Plant Genomes | Bidirectional Mamba | Genome-wide | Fig. 7A,B |

* Because many fungal genomes are compact, taking the region 1 kb 5' of gene starts already covers a significant amount of the genome, including diverse features such as non-coding RNA, coding sequences, regulatory elements, long-terminal repeats and others. ** Several versions trained for different lengths exist.

**Extended Data Table 2 | The block-score discriminates between binding and non-binding sites within sequences with a PWM match**

| PWM Cutoff | n | Class balance | PWM AUROC | Block-Score AUROC |
|---|---|---|---|---|
| 0.001 | 5123635 | 0.018 | 0.784 | 0.813 |
| 0.0001 | 1351010 | 0.047 | 0.678 | 0.799 |
| 0.00001 | 227132 | 0.105 | 0.582 | 0.739 |
| 0.000001 | 38824 | 0.14 | 0.54 | 0.635 |

Area under the ROC curve achieved with the block-score for classifying whether a nucleotide is part of a bound TF motif or not. In difference to Fig. 2d, we restrict only to nucleotides which are part of a PWM match, as selected using different cut-offs of maximum PWM P-value.

# Reporting Summary

## Statistics

For all statistical analyses, confirm that the following items are present in the figure legend, table legend, main text, or Methods section.

| n/a | Confirmed | |
|---|---|---|
| ☐ | ☒ | The exact sample size (*n*) for each experimental group/condition, given as a discrete number and unit of measurement |
| ☒ | ☐ | A statement on whether measurements were taken from distinct samples or whether the same sample was measured repeatedly |
| ☐ | ☒ | The statistical test(s) used AND whether they are one- or two-sided *Only common tests should be described solely by name; describe more complex techniques in the Methods section.* |
| ☒ | ☐ | A description of all covariates tested |
| ☒ | ☐ | A description of any assumptions or corrections, such as tests of normality and adjustment for multiple comparisons |
| ☐ | ☒ | A full description of the statistical parameters including central tendency (e.g. means) or other basic estimates (e.g. regression coefficient) AND variation (e.g. standard deviation) or associated estimates of uncertainty (e.g. confidence intervals) |
| ☐ | ☒ | For null hypothesis testing, the test statistic (e.g. *F*, *t*, *r*) with confidence intervals, effect sizes, degrees of freedom and *P* value noted *Give P values as exact values whenever suitable.* |
| ☒ | ☐ | For Bayesian analysis, information on the choice of priors and Markov chain Monte Carlo settings |
| ☒ | ☐ | For hierarchical and complex designs, identification of the appropriate level for tests and full reporting of outcomes |
| ☐ | ☒ | Estimates of effect sizes (e.g. Cohen's *d*, Pearson's *r*), indicating how they were calculated |

*Our web collection on statistics for biologists contains articles on many of the points above.*

## Software and code

Policy information about availability of computer code

| | |
|---|---|
| Data collection | No software was used for data collection. |
| Data analysis | We used the following  packages: pyRanges (v0.0.127), PyTorch (v2.1.0), biopython (v1.81), flash-attn(2.0.4), matplotlib (v3.7.2), numpy (1.24.4), pandas (2.0.3), plotnine (0.12.3), pyfaidx (0.8.1.1), scikit-learn (1.3.0), scipy (1.10.1), seaborn (1.3.0), transformers (4.26.1), cactus (2.9.3), phast, nhmmer (3.1b2), Clustal-Omega (1.2.4), MAFFT (7.525), ViennaRNA (2.6.4). |
| | For the alignment of DMS-MaPseq reads we used the rf-map module of the RNA Framework and Bowtie2. Count of DMS-induced mutations and coverage and reactivity normalization were performed using the rf-count-genome and rf-norm modules of the RNA Framework. |
| | Code to reproduce all analysis is available at https://github.com/gagneurlab/dependencies_DNALM or at v410.5281/zenodo.16524884 under MIT license. |

For manuscripts utilizing custom algorithms or software that are central to the research but not yet described in published literature, software must be made available to editors and reviewers. We strongly encourage code deposition in a community repository (e.g. GitHub). See the Nature Portfolio guidelines for submitting code & software for further information.

## Data

Policy information about availability of data

All manuscripts must include a data availability statement. This statement should provide the following information, where applicable:

- Accession codes, unique identifiers, or web links for publicly available datasets
- A description of any restrictions on data availability
- For clinical datasets or third party data, please ensure that the statement adheres to our policy

> As stated in the Data availability section of the manuscript: "Data to reproduce the analysis, together with the 69-Saccharomycetales genome alignment and conservation score, as well as the bacterial and archaeal genomes and the plasmid sequences used for the benchmark against RNAalifold, is provided at https://zenodo.org/doi/10.5281/zenodo.12982536. The SpeciesLM models are available at https://huggingface.co/collections/johahi/specieslms-678a39261cfff01c1fa3ae41. Raw DMS-MaPseq data has been deposited to the Gene Expression Omnibus database (GEO), under accession GSE271937 with code gpizuwsexrqfryd ."

## Research involving human participants, their data, or biological material

Policy information about studies with human participants or human data. See also policy information about sex, gender (identity/presentation), and sexual orientation and race, ethnicity and racism.

| | |
|---|---|
| Reporting on sex and gender | N/A |
| Reporting on race, ethnicity, or other socially relevant groupings | N/A |
| Population characteristics | N/A |
| Recruitment | N/A |
| Ethics oversight | N/A |

Note that full information on the approval of the study protocol must also be provided in the manuscript.

# Field-specific reporting

Please select the one below that is the best fit for your research. If you are not sure, read the appropriate sections before making your selection.

☒ Life sciences ☐ Behavioural & social sciences ☐ Ecological, evolutionary & environmental sciences

For a reference copy of the document with all sections, see nature.com/documents/nr-reporting-summary-flat.pdf

# Life sciences study design

All studies must disclose on these points even when the disclosure is negative.

| | |
|---|---|
| Sample size | No sample size calculations were performed as these were predetermined by the datasets we used to analyze the DNA language model. |
| Data exclusions | We did not exclude any data. |
| Replication | Not applicable. |
| Randomization | Not applicable. |
| Blinding | Not applicable. |

# Reporting for specific materials, systems and methods

We require information from authors about some types of materials, experimental systems and methods used in many studies. Here, indicate whether each material, system or method listed is relevant to your study. If you are not sure if a list item applies to your research, read the appropriate section before selecting a response.

## Materials & experimental systems

| n/a | Involved in the study |
|-----|------------------------|
| ☒ | Antibodies |
| ☒ | Eukaryotic cell lines |
| ☒ | Palaeontology and archaeology |
| ☐ | Animals and other organisms (☒) |
| ☒ | Clinical data |
| ☒ | Dual use research of concern |
| ☒ | Plants |

## Methods

| n/a | Involved in the study |
|-----|------------------------|
| ☒ | ChIP-seq |
| ☒ | Flow cytometry |
| ☒ | MRI-based neuroimaging |

# Animals and other research organisms

Policy information about studies involving animals; ARRIVE guidelines recommended for reporting animal research, and Sex and Gender in Research

| Laboratory animals | The goal of this experiment is to show experimental supports complementary to all computational evidence and benchmarks of established RNA structures. To this end we performed DMS for E. coli TOP10 cells at 37C. A single biological replicate was produced. This was deemed sufficient as it showed high correlation to previously published DMS-MaPseq data for the same E. coli strain (GSE247244; r = 0.94, over A/C bases with coverage >= 1,000X). |
|---|---|
| Wild animals | N/A |
| Reporting on sex | N/A |
| Field-collected samples | N/A |
| Ethics oversight | N/A |

Note that full information on the approval of the study protocol must also be provided in the manuscript.

# Plants

| Seed stocks | This study involved no plants. |
|---|---|
| Novel plant genotypes | N/A |
| Authentication | N/A |

