## [Peer Review File · Nature Genetics]

Nucleotide dependency analysis of genomic language models detects functional elements

Corresponding Author: Dr Julien Gagneur

Version 0:

Decision Letter:

12th Dec 2024

Dear Julien,

Firstly, my apologies for the prolonged time in review; we were waiting for a final referee but they got in touch saying they would not be able to submit.

Your Article, "Nucleotide dependency analysis of DNA language models reveals genomic functional elements" has now been seen by 3 referees. You will see from their comments below that while they find your work of interest, some important points are raised. We are interested in the possibility of publishing your study in Nature Genetics, but would like to consider your response to these concerns in the form of a revised manuscript before we make a final decision on publication.

Briefly, the three reviewers all sound supportive of an eventual publication.

Referee #1, coming across as the least convinced currently, states "this will influence research in the field". They have a few comments for additional analysis that seem by and large doable.

Reviewer #2 is straightforwardly positive, with only minor comments.

Referee #3, reflecting their RNA background, is also more supportive but asks for more follow-up on the RNA aspects, especially experimental validation.

In our reading there is a clear path to publication; the computational requests do not seem unduly reasonable, and we think Referee #3's requests for further validation are fair, but given the scope of this study and the overall reviewer positivity we will not editorially require a deep experimental follow-up - though it would be great to add more if that is possible!

To guide the scope of the revisions, the editors discuss the referee reports in detail within the team, including with the chief editor, with a view to identifying key priorities that should be addressed in revision and sometimes overruling referee requests that are deemed beyond the scope of the current study. We hope that you will find the prioritized set of referee points to be useful when revising your study. Please do not hesitate to get in touch if you would like to discuss these issues further.

We therefore invite you to revise your manuscript taking into account all reviewer and editor comments. Please highlight all changes in the manuscript text file. At this stage we will need you to upload a copy of the manuscript in MS Word .docx or similar editable format.

*2) If you have not done so already please begin to revise your manuscript so that it conforms to our Article format instructions, available

http://www.nature.com/ng/authors/article_types/index.html here

*3) Include a revised version of any required Reporting Summary: <https://www.nature.com/documents/nr-reporting-summary.pdf>

Link Redacted

Sincerely,

Michael Fletcher, PhD
Senior Editor, Nature Genetics
ORCID: 0000-0003-1589-7087

Referee expertise:

Referees #1, #2: computational biology; machine learning

Referee #3: computational biology; RNA

Reviewers' Comments:

Reviewer #1 (Remarks to the Author):

Summary

The authors introduce and analyze a method for quantifying nucleotide dependencies in DNA language models trained via self-supervised learning. This approach is particular to models trained this way, and does not apply to the more common supervised training frameworks. Nucleotide dependencies more cleanly reveal regulatory motifs than alternative approaches to scoring nucleotide influence. Although the transcriptional regulation observations are interesting, the method really shines at learning and revealing RNA structure, including subtle pseudoknots that have challenged RNA bioinformatics.

I believe this work will influence research in this field, but it would be helpful to clarify where it most excels, which my major comments aim to address.

Major comments

For several evaluations, including the MPRA, eQTL, and ClinVar, the authors should include a state-of-the-art supervised baseline. In prior evaluations, self-supervised methods generally perform worse than supervised methods. Clearly this approach enhances self-supervised methods, but does it help them catch up to supervised?

In Figure 2D, you match PWM p-values for predicting TF binding, but supervised models for TF binding consistently outperform PWM p-values. Supervised models make use of flanking nucleotides and nearby TF motifs. Why wouldn't the self-supervised models achieve the same? Adding a supervised baseline here would be helpful.

There are some really neat results in the paper where the models with this approach can highlight functionally relevant

nucleotides. However, the authors (very good and insightful) analysis of tandem repeats and the cursory analysis of transposable elements indicates that the models and this approach confound functional conservation and mutational processes. Both of these forces make nucleotides easier for self-supervised learning to predict, and both introduce these nucleotide dependencies. I'd love to see the authors address this tension head on and suggest how one might focus on one or the other to help readers and potential users of the method best make use of it.

Minor comments

For the ClinVar evaluation, it would be helpful to split scores by gene category. I don't understand why your 5' model would make good predictions on the non 5' categories. Does it?

Figure 2C, why wouldn't repeat elements have strong dependencies?

All the switching between models from analysis to analysis and figure to figure is hard to follow. It be helpful if the authors could determine a better way to easily communicate to the reader which model is being used. For example, in the figures, you could include a label or logo indicator of the model used.

Reviewer #1 (Remarks on code availability):

Looks sufficient to reproduce and reimplement.

Reviewer #2 (Remarks to the Author):

The manuscript "Nucleotide dependency analysis of DNA language models reveals genomic functional elements" by Pedro Tomaz da Silva et al. explores how nucleotide dependency maps can uncover functional elements across various biological processes, including transcriptional and post-transcriptional regulation, as well as RNA folding. The study demonstrates that nucleotide dependencies, which the authors introduce and with which they measure the impact of nucleotide substitutions at one position on the likelihood of nucleotides at other positions, are more effective at highlighting functional sequence encodings than "reconstruction probabilities" (i.e. masking the position in the sequence and asking a model for the likelihood of observing the known sequence) or attribution/importance scores (to which I believe it extends too) that are commonly used to interpret DNA language models or deep sequence models (DMs).

This is a beautiful study in terms of interpreting DMs derived from various molecular function sources and across genome complexities. Among its different sections it highlights how nucleotide dependencies are effective at indicating the deleteriousness of a nucleotide and how regulatory elements appear as dense blocks in these maps, facilitating the identification of transcription factor binding sites with an accuracy comparable to experimental models. Additionally, nucleotide dependencies highlight functional interactions in splicing and promoters, and reflect bases in contact within RNA folds, including canonical and non-canonical contacts, pseudoknots, and alternative folding. The study also experimentally validated several novel RNA structural predictions in *E. coli*. In later sections, the study discusses distal interactions and periodic signals like nucleosome occupancy.

Nucleotide dependencies provided a richer visualization of functional relationships revealed by DMs and enable benchmarking of these models, revealing that models trained solely on one species (the human genome) failed to learn the tRNA structures, unlike those trained on multiple species. This highlights the importance of providing sequence perturbations that maintain the sequence function to such models, much like the recently honored discoveries of compensatory mutations as the basis of reconstructing protein folds.

This work is an interesting read, and I think will be of interest to the broader audience of this journal. Further, it provides good food for thought for future developments of DMs.

Some minor comments:

- It would be good to reiterate how the block scores are derived in the legend to Fig. 2 as well as to be clear about which "repeats" are being referenced. Further, it would be good if the nucleotide color schemes would match between the different sequence logo sources.
- Line 210: Only using 7 species of yeast seems too little to derive informative conservation scores. This needs more seq. diversity and species.
- Lines 688f: While I understand that processed files were provided and the source of those should be directly mentioned, please also cite either PMID 31106481 or PMID 31395865 as the actual experimental source of this data.

Reviewer #3 (Remarks to the Author):

This manuscript introduces an impressive approach to uncover functional genomic elements using DNA language models (DNA LMs) through the analysis of nucleotide dependencies. By quantifying how nucleotide substitutions at one position influence probabilities at others, the study generates genome-wide dependency maps, providing new insights into regulatory elements, RNA structures, and the impact of genetic variants. Dependency maps identified transcription factor binding sites, captured RNA secondary and tertiary structures, and outperformed alignment-based conservation methods in assessing the

deleteriousness of variants. The study highlights the superior performance of multispecies-trained DNA LMs compared to single-species models, emphasizing the importance of diverse training data. A few experimental validations confirmed the predictive power of dependency maps in identifying novel RNA structures and functional elements. Overall, I am in favor of this publication as it offers a game-changing tool for computational genomics. However, I do have some additional suggestions for experimental approaches that can improve the study. I have focused the majority of my comments on the RNA section of the manuscript.

1. While the study shows multispecies models outperform single-species ones, it assumes evolutionary conservation across species is the best approach, potentially overlooking species-specific regulatory elements. There are plenty of these examples including emerging cis-regulatory elements in human and chimpanzee specific features such as neural crest development that are perhaps the most interesting in the genome. These limitations should be addressed.
2. With respect to RNA structure predictions, the dependency maps capture secondary and tertiary structures but may miss subtle, context-dependent folding patterns or interactions that are influenced by dynamic cellular conditions. Experimental studies that capture these dynamic states are lacking and should be included. The manuscript as a whole is also lacking adequate experimental to draw many of the conclusions made with respect to RNA structure analysis. The study experimentally validates only a small subset of predicted structures, leaving the broader applicability of the method untested.
3. RNA folding is influenced by the cellular environment, including protein interactions, and kinetic folding pathways. Dependency maps, which analyze static sequence contexts, may not capture these dynamic aspects. While the paper offers an outstanding methodology, its reliance on computational models and limited experimental validation raises questions about generalizability, species-specific applications, and biological relevance in dynamic contexts.
4. While the maps identify pseudoknots and some non-canonical interactions, their accuracy for these features is unclear without broader experimental validation across diverse RNA types. Moreover, training on annotated RNAs could limit discovery of novel RNA structures that deviate from known patterns.
5. The paper emphasizes the advantages of being alignment-free but does not adequately compare its RNA structure predictions with well-established alignment-based methods like RNAalifold, which are gold standards for RNA covariation and structure prediction.
6. The claim that dependency maps outperform alignment-based conservation could be overstated for some genomic regions where alignments offer critical evolutionary insights. I therefore feel that certain parts of the manuscript need to be toned down.

Version 1:

Decision Letter:

Our ref: NG-A66668R

19th Jun 2025

Dear Dr. Gagneur,

Thank you for submitting your revised manuscript "Nucleotide dependency analysis of DNA language models reveals genomic functional elements" (NG-A66668R). It has now been seen by the original referees and their comments are below. The reviewers find that the paper has improved in revision, and therefore we'll be happy in principle to publish it in Nature Genetics, pending minor revisions to satisfy the referees' final requests and to comply with our editorial and formatting guidelines.

Sincerely,

Margot Brandt, PhD
Senior Editor
Nature Genetics
<https://orcid.org/0000-0002-9434-794X>

Reviewer #1 (Remarks to the Author):

The authors have satisfied my concerns and further improved the manuscript.

Reviewer #2 (Remarks to the Author):

Pedro Tomaz da Silva et al. have provided a detailed response to the reviewer comments and produced a revised submission of their manuscript "Nucleotide dependency analysis of DNA language models reveals genomic functional elements". I believe the manuscript has further improved with the edits provided and contains deeper analyses based on some of the reviewer requests. Specifically, I appreciate the additional efforts of creating a deeper yeast conservation score. I must admit that my expectations were more towards the inclusion of a comment about the limited resolution in scores from the small number of species. However, I also appreciate the new result that clarifies that the alignment depth was clearly not the main limitation. The new texts added in revision, especially in the discussion, give a more nuanced reflection on the opportunities and limitations of the dependency maps and the underlying modeling with language models. New method texts regarding the inclusion of Borzoi might need another round of small edits to match the quality of other sections.

Reviewer #2 (Remarks on code availability):

Deposited code is useful, but somewhat minimal in its documentation. Neither GitHub code base or Zenodo has been recently updated (last updates Aug 2024). I am wondering whether updates need to happen to incorporate new analyses and files from the revision.

Reviewer #3 (Remarks to the Author):

I have carefully reviewed the paper and the authors have done a very good job in answering all of my questions.

I believe that the paper is now ready for publication.

Editor

Your Article, "Nucleotide dependency analysis of DNA language models reveals genomic functional elements" has now been seen by 3 referees. You will see from their comments below that while they find your work of interest, some important points are raised. We are interested in the possibility of publishing your study in Nature Genetics, but would like to consider your response to these concerns in the form of a revised manuscript before we make a final decision on publication.

Briefly, the three reviewers all sound supportive of an eventual publication.

Referee #1, coming across as the least convinced currently, states "this will influence research in the field". They have a few comments for additional analysis that seem by and large doable.

Reviewer #2 is straightforwardly positive, with only minor comments.

Referee #3, reflecting their RNA background, is also more supportive but asks for more follow-up on the RNA aspects, especially experimental validation.

In our reading there is a clear path to publication; the computational requests do not seem unduly reasonable, and we think Referee #3's requests for further validation are fair, but given the scope of this study and the overall reviewer positivity we will not editorially require a deep experimental follow-up - though it would be great to add more if that is possible!

To guide the scope of the revisions, the editors discuss the referee reports in detail within the team, including with the chief editor, with a view to identifying key priorities that should be addressed in revision and sometimes overruling referee requests that are deemed beyond the scope of the current study. We hope that you will find the prioritized set of referee points to be useful when revising your study. Please do not hesitate to get in touch if you would like to discuss these issues further.

We therefore invite you to revise your manuscript taking into account all reviewer and editor comments. Please highlight all changes in the manuscript text file. At this stage we will need you to upload a copy of the manuscript in MS Word .docx or similar editable format.

*2) If you have not done so already please begin to revise your manuscript so that it conforms to our Article format instructions, available here.

***3) Include a revised version of any required Reporting Summary:**

Please be aware of our guidelines on digital image standards.

<https://mts-ng.nature.com/cgi-bin/main.plex?el=A2G1BhN7A2Bqmm6J1A9ftdLGYEnnVQC2toAQbxkXwgZ>

We have addressed all comments in a point-by-point response below. We have further noticed a minor mistake in figure 5B and corrected it.

Reviewer #1 (Remarks to the Author):

Summary

The authors introduce and analyze a method for quantifying nucleotide dependencies in DNA language models trained via self-supervised learning. This approach is particular to models trained this way, and does not apply to the more common supervised training frameworks. Nucleotide dependencies more cleanly reveal regulatory motifs than alternative approaches to scoring nucleotide influence. Although the transcriptional regulation observations are interesting, the method really

shines at learning and revealing RNA structure, including subtle pseudoknots that have challenged RNA bioinformatics.

I believe this work will influence research in this field, but it would be helpful to clarify where it most excels, which my major comments aim to address.

Major comments

For several evaluations, including the MPRA, eQTL, and ClinVar, the authors should include a state-of-the-art supervised baseline. In prior evaluations, self-supervised methods generally perform worse than supervised methods. Clearly this approach enhances self-supervised methods, but does it help them catch up to supervised?

We have added a strong supervised baseline, Borzoi, a state-of-the-art supervised model for gene expression ¹, and updated Fig 1C (gene expression in a promoter mutagenesis screen) as well as supplementary figures (Fig S1C,F, eQTL and ClinVar non-coding). These analyses show that the variant effect score performs reasonably well despite not being trained for those tasks. Moreover, fitting a linear model on the influence score and Borzoi score per promoter (mutagenesis screen) on those data indicates that the dependency scores, and therefore the underlying DNA LMs, entail complementary information to Borzoi's predicted effects. This observation is interesting for the computational regulatory genomics field.

This being said, it was not the goal of this benchmark to deliver a fully fleshed-out variant effect score that can rival the performance of state-of-the-art supervised models, particularly on expression prediction in mammals where ENCODE provides an extremely large amount of labeled data. Our stated goal with this benchmark was to show that dependencies capture functional nucleotides, and they do so better than the language model probabilities. There are many other ways to score variants using DNA-LM and possibly many modifications that can be made to increase their predictive power on a specific downstream task.

Moreover, the influence score and scores offered by supervised models measure fundamentally different things. The influence score is agnostic to the tissue a variant acts in, and to the biological mechanism it impacts which can include anything under selection (transcription, splicing, translation, DNA replication, etc.). Borzoi, meanwhile, is tissue-specific, is focused on particular molecular outcomes (mostly transcriptional), and predicts the (signed!) impact of a variant on gene expression.

Altogether, while building a fully fleshed-out gene expression predictor based on a DNA-LM goes beyond the scope of this study, the new analyses on these three evaluations indicate that this research direction is promising and that dependency analysis can help assess the potential added value of DNA-LMs.

In Figure 2D, you match PWM p-values for predicting TF binding, but supervised models for TF binding consistently outperform PWM p-values. Supervised models make use of flanking nucleotides and nearby TF motifs. Why wouldn't the self-supervised models achieve the same? Adding a supervised baseline here would be helpful.

Unfortunately, this analysis was done in yeast, where - to our knowledge - there is no supervised model of TF binding that covers all the factors we benchmark on. Thus, accurately satisfying this request would require designing, training and optimizing a unified model of TF binding in yeast, which we believe is out of the scope of this paper.

This being said, we agree the underlying point (whether the DNA LMs capture motif context) is important and we thank the reviewer for drawing attention to this. We can evaluate whether the self-supervised model takes into account the context of the motif by framing the benchmark slightly differently. Specifically, rather than performing it genome-wide, we restrict the analysis only to nucleotides that form part of a PWM-match in the first place, using different PWM P-value significance cutoffs to stratify between weaker and stronger PWM matches (see below). We then tested the ability of the PWM P-value to differentiate binding from non-binding motif instances within each strata. While the PWM P-value performance degrades with increasingly stringent cutoffs, the block score remains predictive. This implies that the prediction of the self-supervised model block score cannot be based solely on the sequence in the motif itself.

PWM Cutoff	n	Class balance	PWM AUROC	Block-Score AUROC
0.001	5,123,635	0.018	0.784	0.813
0.0001	1,351,010	0.047	0.678	0.799
0.00001	227,132	0.105	0.582	0.739
0.000001	38,824	0.14	0.54	0.635

We have included this analysis in the supplementary materials and mentioned it in the Results section.

In conclusion, these analyses show that the self-supervised model makes use of the sequence context of the motif. However, as mentioned in the manuscript, the remarkable result here is that the model was able to identify those motifs without any experimental data, and therefore has a strong potential for studying domains with scarcer experimental data such as post-transcriptional regulation and non-model species.

There are some really neat results in the paper where the models with this approach can highlight functionally relevant nucleotides. However, the authors (very good and insightful) analysis of tandem repeats and the cursory analysis of transposable elements indicates that the models and this approach confound functional conservation and mutational processes. Both of these forces make nucleotides easier for self-supervised learning to predict, and both introduce these nucleotide dependencies. I'd love to see the authors address this tension head on and suggest how one might focus on one or the other to help readers and potential users of the method best make use of it.

We thank the reviewer for the positive feedback. As pointed out, the language model predictions are driven by signals stemming from functional conservation, but also from mutational biases and easy-to-predict low-complexity regions that follow simple rules such as repeats. It is hard to establish robust guidelines for users to distinguish which predictions are more driven by functional conservation from the others within the scope of this paper. However, we have expanded the discussion on this point hinting towards two possible directions.

On the one hand, post-processing techniques could be proposed in the future to distinguish bases whose reconstruction is driven by mutational processes from those driven by selection. One can imagine using repeat filters for instance. On the other hand, the DNA LMs themselves could be trained differently, aiming explicitly at modeling mutational processes. One step in this direction was proposed by Benegas et al. ² who down-weighted repeat elements during training of the DNA LM. Further approaches should be explored to also cope with other mutational biases.

Either way, we believe that dependencies will be a useful tool to analyze these different strategies and detect possible biases, as they allow to systematically probe DNA-LM predictions. We have now included those points in the discussion.

Minor comments

For the ClinVar evaluation, it would be helpful to split scores by gene category. I don't understand why your 5' model would make good predictions on the non 5' categories. Does it?

Stratifying the non-coding pathogenic variants by category shows that 92% are variants affecting splicing. 5' UTRs are often spliced (about one third of the human 5'UTR are spliced ³ giving to our 5' model, which comprises the 2 kb 5' of the start codons, the possibility to recognize splice sites. Intriguingly, we find robust discriminative performance across the ClinVar gene location category, although we cannot exclude that many of those (e.g. the intronic variants not directly on the splice sites) are not related to splicing as well.

As we mentioned in the manuscript, ClinVar is heavily biased toward variants that were captured by manual annotations and classical tools including splicing models (MaxEntscan, etc). Although looking favorable, we opted to not integrate this analysis to not misleadingly emphasize this benchmark.

Figure 2C, why wouldn't repeat elements have strong dependencies?

This point was perhaps unclear. We did not want to convey the message that repeats do not have dependencies. There are various types of repeat elements and repetitive elements. While DNA LMs capture them well, i.e. reconstruct their sequences well, they do not all end up showing strong block dependencies. We show a poly-oligomer (polyA) example in Fig 2A. Mutating a single A has little influence in reconstructing the others, although the model likely uses the overall large number of As to make its predictions in this region. We show tandem repeats in Fig 5, demonstrating that DNA LMs capture those as local copies and therefore do not exhibit block dependencies. Other repeats include transposons and self-replicating elements that are usually large and for which individual dependencies tend also to be weaker than in short TF motifs. Altogether, while dependencies are found within repeats, we find that repeats show weaker block scores than TF motifs.

We have added a sentence in the paragraph referring to Figs. 2A,B to make this point clearer.

All the switching between models from analysis to analysis and figure to figure is hard to follow. It be helpful if the authors could determine a better way to easily communicate to the reader which model is being used. For example, in the figures, you could include a label or logo indicator of the model used.

Thanks for the suggestion. We have added a banner to the panels in Figures where different models are used. For the remaining figures, we state in the legend which model is used throughout.

Reviewer #1 (Remarks on code availability):

Looks sufficient to reproduce and reimplement.

Reviewer #2 (Remarks to the Author):

The manuscript "Nucleotide dependency analysis of DNA language models reveals genomic functional elements" by Pedro Tomaz da Silva et al. explores how nucleotide dependency maps can uncover functional elements across various biological processes, including transcriptional and post-transcriptional regulation, as well as RNA folding. The study demonstrates that nucleotide dependencies, which the authors introduce and with which they measure the impact of nucleotide substitutions at one position on the likelihood of nucleotides at other positions, are more effective at highlighting functional sequence encodings than "reconstruction probabilities" (i.e. masking the position in the sequence and asking a model for the likelihood of observing the known sequence) or attribution/importance scores (to which I believe it extends too) that are commonly used to interpret DNA language models or deep sequence models (DMs).

This is a beautiful study in terms of interpreting DMs derived from various molecular function sources and across genome complexities. Among its different sections it highlights how nucleotide dependencies are effective at indicating the deleteriousness of a nucleotide and how regulatory elements appear as dense blocks in these maps, facilitating the identification of transcription factor binding sites with an accuracy comparable to experimental models. Additionally, nucleotide dependencies highlight functional interactions in splicing and promoters, and reflect bases in contact within RNA folds, including canonical and non-canonical contacts, pseudoknots, and alternative folding. The study also experimentally validated several novel RNA structural predictions in *E. coli*. In later sections, the study discusses distal interactions and periodic signals like nucleosome occupancy.

Nucleotide dependencies provided a richer visualization of functional relationships revealed by DMs and enable benchmarking of these models, revealing that models trained solely on one species (the human genome) failed to learn the tRNA structures, unlike those trained on multiple species. This highlights the importance of providing sequence perturbations that maintain the sequence function to such models, much like the recently honored discoveries of compensatory mutations as the basis of reconstructing protein folds.

This work is an interesting read, and I think will be of interest to the broader audience of this journal. Further, it provides good food for thought for future developments of DMs.

Some minor comments:

- It would be good to reiterate how the block scores are derived in the legend to Fig. 2 as well as to be clear about which "repeats" are being referenced. Further, it would be good if the nucleotide color schemes would match between the different sequence logo sources.

We have now clarified that RepeatMasker is used to define repeats (Fig. 2C legend and main text), as well as mentioned in the main text the categories of repeats that can be identified in *S. cerevisiae*, and reiterated the definition of block score in the caption. We have also updated the color scheme to be consistent between logo sources.

- Line 210: Only using 7 species of yeast seems too little to derive informative conservation scores. This needs more seq. diversity and species.

We used the widely used 7-way phastCons score from UCSC, thereby adhering to the definition of conservation widely used in practice in the yeast community. We agree that for the purpose of comparing the methods more fairly, generating alignment-based conservation measures using a larger set of yeast genomes is meaningful. However, to our knowledge, no such scores are currently available.

Nevertheless, we attempted to address this point by computing conservation scores based on alignments for more species. To this end, we used progressive cactus to align 69 Saccharomycetales species using default parameters and specifying *S. cerevisiae* as a reference quality genome. We then extracted four-fold degenerate sites and used phyloFit with the EM algorithm to estimate a neutral model. Using this neutral model and the alignment, we ran phastCons with the parameters `--rho 0.3 --estimate-rho --target-coverage 0.4 --expected-length 23`, which are identical to the ones used in the 7-way alignment. We also ran phyloP, with `--method LRT --mode CONACC`.

The resulting conservation scores differed yet significantly correlated with the 7-way phastCons scores ($\rho = 0.65$ for phastCons, 0.51 for phyloP). These conservation scores did not perform much better in our benchmark (new Supplementary Fig. 2). Whether this is because of the quick evolution in the yeast lineage - which makes it difficult to align non-coding sequences beyond the genus boundary - or because our lack in expertise in this area leads to us using default but suboptimal parameter choices is something we cannot confidently say.

This being said, part of the reason for conservation to perform so badly on the motif discrimination task is that many of the 5' regions partially overlap other coding sequences. Block score and PWM scanning are not noticeably impacted by this, but conservation scores generally are highest in coding regions, making them ill-suited to finding motifs in this setup. Explicitly removing all non-motif-binding coding sequences increases the discriminative power of conservation scores, although they still perform much worse than block-score and PWM scanning. (see new Supplementary Fig. 2). We have added a note on this to the main text.

- Lines 688f: While I understand that processed files were provided and the source of those should be directly mentioned, please also cite either PMID 31106481 or PMID 31395865 as the actual experimental source of this data.

We thank the reviewer for pointing this out. We have now cited the data accordingly in the results and methods.

Reviewer #3 (Remarks to the Author):

This manuscript introduces an impressive approach to uncover functional genomic elements using DNA language models (DNA LMs) through the analysis of nucleotide dependencies. By quantifying how nucleotide substitutions at one position influence probabilities at others, the study generates genome-wide dependency maps, providing new insights into regulatory elements, RNA structures, and the impact of genetic variants. Dependency maps identified transcription factor binding sites, captured RNA secondary and tertiary structures, and outperformed alignment-based conservation methods in assessing the deleteriousness of variants. The study highlights the superior performance of multispecies-trained DNA LMs compared to single-species models, emphasizing the importance of diverse training data. A few experimental validations confirmed the predictive power of dependency maps in identifying novel RNA structures and functional elements. Overall, I am in favor of this publication as it offers a game-changing tool for computational genomics. However, I do have some additional suggestions for experimental approaches that can improve the study. I have focused the majority of my comments on the RNA section of the manuscript.

1. While the study shows multispecies models outperform single-species ones, it assumes evolutionary conservation across species is the best approach, potentially overlooking species-specific regulatory elements. There are plenty of these examples including emerging cis-regulatory elements in human and chimpanzee specific features such as neural crest development that are perhaps the most interesting in the genome. These limitations should be addressed.

We agree that it is possible to imagine scenarios of evolutionary novelty that will be very difficult to learn for any DNA-LM, single or multi-species. To clarify this, we now discuss how DNA-LMs relate to yet distinguish from homology-based and alignment-based approaches (new second paragraph of the discussion).

Unlike alignment-based approaches, multispecies DNA-LMs do not strictly rely on conservation of ancestral sequence across species, but mostly leverage statistical occurrence across genes and species. Suppose a regulatory motif of a conserved transcription factor evolves to be at a novel functional position in the genome (e.g. in a promoter and enhancer). In that case, a multi-species DNA LM can in theory still leverage its overrepresentation in similar sequence contexts across non-homologous sequences both within and across species to capture it.

Our metazoan model, which has been trained only in the sequences flanking the start codon, is not ideal to analyse the type of emerging enhancers referred to. In our previous study, we showed that we could recover promoter architecture and TF binding sites in *S. cerevisiae* with a model that has not seen any of the species of the *saccharomyces* genus, too far for most of these motifs to be captured by alignments⁴. There is other evidence indicating that multispecies DNA-LMs can capture species-specific elements. For instance, our SpeciesLM Metazoa model will often reconstruct Alu elements with high confidence (see example below in the vicinity of one of the Kircher et al. promoters), even though they only appear in a small minority of species the model was trained on. Altogether, there is no a priori reason why an element that is sufficiently overrepresented in a single genome cannot also be detectable by a multispecies DNA-LM.

2. With respect to RNA structure predictions, the dependency maps capture secondary and tertiary structures but may miss subtle, context-dependent folding

patterns or interactions that are influenced by dynamic cellular conditions. Experimental studies that capture these dynamic states are lacking and should be included.

We appreciate the reviewer's feedback but find ourselves unclear about the specific concern being raised.

If the question is about capturing interactions that only occur in specific conditions, then in principle such interactions should be captured if they are functionally important and therefore reflected through co-evolution. By itself, the model cannot infer under which conditions those interactions take place. This is similar to TF binding sites which can only be functional under conditions for which the cognate transcription factor is expressed.

If the question is about capturing multiple conformations for a single RNA, this may in fact be one aspect where dependency maps, which allow individual nucleotides to show interactions with multiple nucleotides, are beneficial compared to methods searching for a single optimal structure. We show examples of alternative RNA conformations for the Tryptophan operon (Fig. 4F), for which tryptophan abundance regulates the switch between terminator and antiterminator conformations.

We have added a statement to the 7th paragraph of the discussion. We hope this edit and this response give a clearer picture of the capability of our method in this regard.

The manuscript as a whole is also lacking adequate experimental to draw many of the conclusions made with respect to RNA structure analysis. The study experimentally validates only a small subset of predicted structures, leaving the broader applicability of the method untested.

We had provided a systematic evaluation against RNA structure databases (which represent a large-scale evaluation since the self-supervised models are not trained at predicting structures), and the experimental validation of 4 new structures in *E. coli* with a DMS assay. We acknowledge that the assessment of the dependencies against the entire new DMS measurements transcriptome-wide was lacking.

To address this, we have now systematically scanned dependency maps for anti-parallel dependencies using convolutional filters for all non-coding regions in *E. coli*. As done for the search for novel RNA structures in *E. coli* (Methods), we considered positions in the dependency map with a convolution value of at least 25 as part of an anti-parallel pattern and therefore belonging to a predicted structural contact. Based on this, we then derived whether a nucleotide belongs to any predicted structural contact. We then performed two analyses. In the new supplementary Figure S4G we show the distribution of DMS mutation frequencies for all A and C nucleotides (the nucleotides probed by DMS) in *E. coli* non-coding regions predicted to be in a structured contact vs all remaining non-coding nucleotides. Nucleotides predicted to be in a structured contact have significantly lower mutation rates ($P < 10^{-16}$, Wilcoxon Rank-sum test), which indicates they were more protected from DMS and therefore more likely to be in Watson-crick base-pairing. We next leveraged DMS data to perform experimentally-constrained RNA secondary structure prediction using the RNA Framework and the Vienna RNA package, and derived base-pairing probabilities

for all bases. In supplementary S4H we show that the probability of pairs predicted by our dependency maps is significantly higher than that expected for all possible pairs ($P < 10^{-16}$, Wilcoxon Rank-sum test). Altogether, these analyses show that, overall, the structural contacts predicted by the anti-parallel patterns in dependency maps can efficiently capture experimentally measured RNA base-pair contacts transcriptome-wide.

3. RNA folding is influenced by the cellular environment, including protein interactions, and kinetic folding pathways. Dependency maps, which analyze static sequence contexts, may not capture these dynamic aspects. While the paper offers an outstanding methodology, its reliance on computational models and limited experimental validation raises questions about generalizability, species-specific applications, and biological relevance in dynamic contexts.

We thank the reviewer for the positive feedback on the methodology. We hope that doubts regarding the generalizability and broad experimental validation can be mitigated by our newly added genome-wide benchmark of predicted structural contacts in the non-coding genome of *E. coli*, the benchmarks we had already performed on multiple known RNA structures across different species and RNA families with evaluation of non-canonical RNA contacts, and a further systematic evaluation of pseudoknots shown in the next answer.

We agree that the dynamic aspects of RNA structure in the context of different environments are not fully captured by dependency maps. However, this is not a drawback specific to our approach. Computational methods based on sequence alignment and coevolution (e.g. RNAalifold) or state-of-the-art protein folding methods like AlphaFold predict structural contacts in RNA and proteins which are not specific to the cellular environment but have proven extremely useful. Compared to previous computational methods in RNA, dependencies go a step further by also highlighting alternative structures. However, revealing in which experimental conditions these are used will require condition-specific experimental data.

We hope that the new analyses along with the extension of the Discussion are addressing this comment. We agree with the reviewer that the full scope of applications of dependency maps extends beyond the content of this work, and are looking forward to the community to continue leveraging its potential.

4. While the maps identify pseudoknots and some non-canonical interactions, their accuracy for these features is unclear without broader experimental validation across diverse RNA types.

The ability of dependencies to identify pseudoknots lacked a systematic evaluation in our original manuscript. To address this we first downloaded the dataset bpRNA-1m(90)⁵ which contains 28,370 annotated RNA structures with less than 90% sequence similarity obtained from the databases CRW, tmRNA, SRP, tRNAdb2009, RNP, RFAM and PDB. From these, we extracted all structures that contain pseudoknot contacts and are no longer than 1,024 nt (the maximum context length the DNA LM RiNALMo has been trained on). These resulted in 2,530 structures of varying lengths and sources. We then computed the dependency maps for each one of these structures and evaluated their ability to predict whether a pair of

nucleotides belongs to a pseudoknot contact (positive set) or does not belong to a structure contact (pseudoknot or canonical structure contact - negative set). We found that dependencies generally capture pseudoknot contacts (new Supplementary Fig 4F).

An example of a dependency map (left panel), the ground truth annotation of pseudoknots (right panel) and ground truth annotation of the whole structure (bottom panel) is provided below.

As for other non-canonical contacts, we had benchmarked non Watson-Crick or Wobble contacts from the CompaRNA database spanning 201 RNAs, which is based on RNA structures available in the PDB database (Fig 4B-D, already in the first submission). Unfortunately, obtaining RNA structures at high throughput is not feasible, explaining the scarcity of ground truth in the field for tertiary structure contacts. Sequence-based high-throughput assays are also limited in this regard. DMS specifically modifies unpaired adenines and cytosines *in vivo* at their Watson-Crick base-pairing positions and therefore does not allow us to probe non Watson-Crick base-pairing genome-wide. Therefore, we hope to convince this reviewer that the current version of the manuscript already evaluates as much as feasible non-canonical contacts of experimentally determined RNA structures available to date.

Moreover, training on annotated RNAs could limit discovery of novel RNA structures that deviate from known patterns.

In contrast to supervised models, DNA Language models can be trained on any available sequence regardless of whether the sequence is annotated. The DNA language model RiNALMo which we use for the RNA structure analyses, has been pretrained on 36 million non-coding sequences of both annotated and unannotated RNA sequences from RNACentral, nt, Rfam and Ensembl spanning a wide variety of species. We are now including in the text which sequences have been used for the training of this model. Furthermore, DNA Language models differ from classical approaches based on sequence alignment as they can capture patterns across non-homologous sequences. If a particular RNA sequence was not in the pretraining of the model, the patterns it has learned from other sequences can still be applicable on an unseen RNA. This allows DNA LMs and, in particular, the dependencies extracted from them to reveal novel structures. Nonetheless, the choice of sequences, whether annotated or not, has a decisive role on which sequence elements models can capture as we show in the last section of the results (Fig. 7) and further note in the discussion section.

5. The paper emphasizes the advantages of being alignment-free but does not adequately compare its RNA structure predictions with well-established alignment-based methods like RNAalifold, which are gold standards for RNA covariation and structure prediction.

While training a new state-of-the-art model for RNA structure prediction is beyond the scope of this study, we understand the interest in providing a side-by-side comparison to gold standard structural prediction models to better characterize qualitatively and quantitatively the potential of DNA-LMs and dependency analyses.

To this end, we now compare dependencies against i) the gold-standard alignment-based model RNAalifold and ii) fine-tuned RiNALMo. This comparison allows distinguishing the switch from alignment-based approach (RNAalifold) to an LLM-based, alignment-free approach (fine-tuned RiNALMo) and from secondary structure prediction (the two latter) against general nucleotide interactions (dependency analysis of the non-fine-tuned LLM underpinning the fine-tuned RiNALMo).

We performed this benchmark against experimentally derived structures from PDB annotated in the compaRNA database ⁶. We avoided other RNA structure databases as they mostly contain RNA structures inferred from alignment-based covariation analysis approaches like RNAalifold. This highly limits the available number of structures but allows for a fairer benchmark.

We considered two settings. In one setting, we use the alignments obtained from the RFAM database entry corresponding to the PDB entry. These are high-quality manually curated seed alignments, representing the best-case scenario for RNAalifold. We found that RNAalifold and the fine-tuned RiNALMo perform on par at predicting canonical secondary structure contacts (cis Watson-crick or wobble) (Fig. S4D for the RNAalifold run with

RIBOSUM scoring, which leads to a better performance, and the figure below for the RNAalifold run with the default energy model). These results are consistent with the original report of the Rinalmo's authors and recent work by other labs showing the performance of alignment-free LLM-based approaches for secondary structure prediction (e.g. RNA-FM ⁷). We note, however, that the RNAalifold performance dropped when using a non-manually curated alignment with the same sequences, be it either Clustal-Omega (v1.2.4) ⁸ or MAFFT (v7.525) ⁹. In comparison, the raw dependencies do not perform as well as RNAalifold and the fine-tuned Rinalmo. For non-canonical contacts, we observed that the dependencies captured contacts missed by the fine-tuned Rinalmo and RNAalifold.

In the second setting, we searched for sequences on a database of 220,478 bacterial and archaeal genomes and plasmids downloaded from NCBI to construct an alignment (Methods). In contrast to the first setting, this setting more realistically reflects the situation where there is no manual curation of alignment and avoids use of prior knowledge about the structure in searching for the sequences and defining the alignment. Here nucleotide dependencies perform closer to RNAalifold and are able to find non-canonical contacts missed both by the fine-tuned RiNALMo and RNAalifold.

In conclusion, alignment-free methods are comparable or superior in performance to RNAalifold as shown by the performance of the fine-tuned RiNALMo. While nucleotide dependencies generally perform worse for secondary structure contacts than RNAalifold especially from manually curated alignments, they provide non-canonical interactions missed by both RNAalifold and the fine-tuned RiNALMo.

6. The claim that dependency maps outperform alignment-based conservation could be overstated for some genomic regions where alignments offer critical evolutionary insights. I therefore feel that certain parts of the manuscript need to be toned down.

We share the view that the conceptual distinction between alignment-based analyses, which aim at comparing sequences deriving from a common ancestor sequence, provides evolutionary insights that are not replaced by DNA-LMs which, instead, capture statistical patterns within and between orthologs sequences. We have made several edits in the discussion to clarify this, notably the new second paragraph. We believe that these edits make an important clarification and we thank the reviewer for stressing this point.

References

1. Linder, J., Srivastava, D., Yuan, H., Agarwal, V. & Kelley, D. R. Predicting RNA-seq coverage from DNA sequence as a unifying model of gene regulation. *Nat. Genet.* 1–13 (2025) doi:10.1038/s41588-024-02053-6.
2. Benegas, G., Batra, S. S. & Song, Y. S. DNA language models are powerful predictors of genome-wide variant effects. *Proc. Natl. Acad. Sci.* **120**, e2311219120 (2023).
3. Cenik, C., Derti, A., Mellor, J. C., Berriz, G. F. & Roth, F. P. Genome-wide functional

- analysis of human 5' untranslated region introns. *Genome Biol.* **11**, R29 (2010).
4. Karollus, A. *et al.* Species-aware DNA language models capture regulatory elements and their evolution. *Genome Biol.* **25**, 83 (2024).
 5. Danaee, P. *et al.* bpRNA: large-scale automated annotation and analysis of RNA secondary structure. *Nucleic Acids Res.* **46**, 5381–5394 (2018).
 6. Puton, T., Kozlowski, L. P., Rother, K. M. & Bujnicki, J. M. CompaRNA: a server for continuous benchmarking of automated methods for RNA secondary structure prediction. *Nucleic Acids Res.* **41**, 4307–4323 (2013).
 7. Chen, J. *et al.* Interpretable RNA Foundation Model from Unannotated Data for Highly Accurate RNA Structure and Function Predictions. Preprint at <https://doi.org/10.48550/arXiv.2204.00300> (2022).
 8. Sievers, F. & Higgins, D. G. Clustal Omega for making accurate alignments of many protein sequences. *Protein Sci.* **27**, 135–145 (2018).
 9. Katoh, K., Rozewicki, J. & Yamada, K. D. MAFFT online service: multiple sequence alignment, interactive sequence choice and visualization. *Brief. Bioinform.* **20**, 1160–1166 (2019).

Reviewer #1 (Remarks to the Author):

The authors have satisfied my concerns and further improved the manuscript.

Reviewer #2 (Remarks to the Author):

Pedro Tomaz da Silva et al. have provided a detailed response to the reviewer comments and produced a revised submission of their manuscript “Nucleotide dependency analysis of DNA language models reveals genomic functional elements”. I believe the manuscript has further improved with the edits provided and contains deeper analyses based on some of the reviewer requests. Specifically, I appreciate the additional efforts of creating a deeper yeast conservation score. I must admit that my expectations were more towards the inclusion of a comment about the limited resolution in scores from the small number of species. However, I also appreciate the new result that clarifies that the alignment depth was clearly not the main limitation. The new texts added in revision, especially in the discussion, give a more nuanced reflection on the opportunities and limitations of the dependency maps and the underlying modeling with language models. New method texts regarding the inclusion of Borzoi might need another round of small edits to match the quality of other sections.

We have now edited the methods text regarding Borzoi to improve readability.

Reviewer #2 (Remarks on code availability):

Deposited code is useful, but somewhat minimal in its documentation. Neither GitHub code base or Zenodo has been recently updated (last updates Aug 2024). I am wondering whether updates need to happen to incorporate new analyses and files from the revision.

We have updated the code on March 13th to include the results after the revisions, hardware and software requirements and updates to the quickstart notebook to compute gLM dependencies and variant influence scores. We also made our language models SpeciesLM Metazoa and Fungi available on huggingface to increase ease of use.

Reviewer #3 (Remarks to the Author):

I have carefully reviewed the paper and the authors have done a very good job in answering all of my questions.

I believe that the paper is now ready for publication.